# SARS-CoV-2 genomic and subgenomic RNAs in diagnostic samples are not an indicator of active replication

Soren Alexandersen [1,2,3 ✉], Anthony Chamings [1,2] & Tarka Raj Bhatta [1,2]

Severe acute respiratory syndrome coronavirus-2 (SARS-CoV-2) was first detected in late December 2019 and has spread worldwide. Coronaviruses are enveloped, positive sense, single-stranded RNA viruses and employ a complicated pattern of virus genome length RNA replication as well as transcription of genome length and leader containing subgenomic RNAs. Although not fully understood, both replication and transcription are thought to take place in so-called double-membrane vesicles in the cytoplasm of infected cells. Here we show detection of SARS-CoV-2 subgenomic RNAs in diagnostic samples up to 17 days after initial detection of infection and provide evidence for their nuclease resistance and protection by cellular membranes suggesting that detection of subgenomic RNAs in such samples may not be a suitable indicator of active coronavirus replication/infection.

[1] Geelong Centre for Emerging Infectious Diseases, Geelong, VIC 3220, Australia. [2] Deakin University, Geelong, VIC 3220, Australia. [3] Barwon Health, University Hospital Geelong, Geelong, VIC 3220, Australia. ✉email: soren.alexandersen@deakin.edu.au

Human coronavirus disease 2019 (COVID-19) emerged in late December 2019[1,2] and a novel betacoronavirus, subsequently named severe acute respiratory syndrome coronavirus-2 (SARS-CoV-2), shown to be the cause. This virus could rather easily transmit from person to person and rapidly spread worldwide[3,4]. SARS-CoV-2 belongs to the Order *Nidovirales*, Family *Coronaviridae*, Subfamily *Orthocoronavirinae*, Genus *Betacoronavirus*, Subgenus *Sarbecovirus*, Species *Severe acute respiratory syndrome-related coronavirus* and individuum SARS-CoV-2 with the addition of the strain/sequence, e.g., SARS-CoV-2 Wuhan-Hu-1 as the reference strain[5].

Similar to other coronaviruses, SARS-CoV-2 is an enveloped, positive sense, single stranded RNA virus with a genome of nearly 30,000 nucleotides[6]. After having entered the host cell, replication of coronaviruses initially involves generation of a complementary negative sense genome length RNA for amplification of plus strand virus genome RNA, as well as transcription of a series of plus strand subgenomic RNAs all with a common leader joined to gene sequences in the 3′-end of the virus genome. Virus replication and transcription both involve cytoplasmic membrane structures forming virus replication/transcription organelles. These structures include virus proteins derived from proteolytic processing of the polyprotein encoded in the 5′ two thirds of the virus genome (termed Open Reading Frame (Orf) 1a and 1b) with a minus 1 ribosomal frameshift between Orf1a and 1b, and translated from the full length plus sense virus genome RNA. A set of subgenomic RNAs are also generated, most likely from a complex mechanism involving paused negative sense RNA synthesis leading to a nested set of negative sense RNAs from the 3′end of the virus genome joined to a common 5′-leader sequence of approximately 70 nucleotides[7,8]. The pausing of the virus replication/transcription complex occurs at so-called transcription-regulatory sequences (TRS) located immediately adjacent to open reading frames for these virus genes[9,10]. These nested negative sense RNAs in turn serve as templates for transcription of plus strands able to serve as a nested set of virus mRNAs for translation of specific proteins from the 3′-third of the virus genome[7]. These subgenomic mRNAs of SARS-CoV-2, as illustrated in Kim et al.[9], are thought to encode the following virus proteins: structural proteins spike (S), envelope (E), membrane (M) and nucleocapsid protein (N) and several accessory proteins for SARS-CoV-2 thought to include 3a, 6, 7a, 7b, 8, and 10[9]. Furthermore, it appears that the expression of the N protein is required for efficient coronavirus subgenomic mRNA transcription[7].

The subcellular site/s of coronavirus RNA replication and transcription in the cytoplasm of infected cells is not fully defined, but thought to involve so-called "double-membrane vesicles" (DMV) in or on, which the virus replication complex synthesize the needed double and single stranded full length genomic and subgenomic RNAs[7,8,11]. While it is still unclear whether this RNA synthesis takes place inside or on the outside of these vesicles, it is thought that the membranes somehow "protect" the synthesized RNA, including double stranded RNA, from host cell recognition and response, and also from experimental exposure to RNase[8,12]. In addition, it has been shown that coronavirus cytosolic RNA is protected from so-called "nonsense-mediated decay" (NMD) by the virus N protein and thus are more stable in that environment compared to what would have been expected for nonspliced RNA[13].

While it was originally thought that coronavirus virions contained subgenomic RNAs in addition to the virus plus strand genomic length RNA, it has now been shown that these subgenomic RNAs do not contain a packaging signal and are not found in highly purified, cellular membrane free, coronavirus virions[14]. However, it is important to stress, that unless specific steps to remove cellular membranes are used for sample preparation and virion purification, such subgenomic coronavirus RNAs are tightly associated with membrane structures, and less purified coronavirus preparations are well known to include subgenomic RNAs that, similar to virion RNA, are nuclease resistant[15].

One study has been published looking at the abundance of subgenomic RNAs for SARS-CoV-2 cultured in Vero cells[9]. That study indicated that while the predicted spike (S; Orf2), Orf3a, envelope (E; Orf4), membrane (M; Orf5), Orf6, Orf7a, and nucleocapsid protein (N; Orf9) subgenomic RNAs were found at high levels in cell culture, only low levels of the Orf7b subgenomic RNA was detected and the Orf10 subgenomic RNA (also sometimes referred to as Orf15[10]) was detected at extremely low level (1 read detected, corresponding to only 0.000009% of reads analysed)[9]. This far, little has been published in regards to the presence of SARS-CoV-2 subgenomic RNAs in samples from infected people. A single study by Wölfel et al.[16], looked specifically for the presence of the E gene subgenomic RNA by a PCR and took the presence of subgenomic RNA as an indication of active virus infection/transcription. That study could detect E gene subgenomic RNA at a level of only 0.4% of the virus genome RNA in sputum samples from days 4–9 of infection, but only up to day 5 in throat swab samples[16]. That study assumed a correlation between the presence of the subgenomic E mRNA and active virus replication/transcription and thus active infection, however, this assumption may not be accurate considering what has been mentioned above about the membrane associated nature of coronavirus RNA and their stability/protection from the host cell response and from RNases.

In this work we describe the detection of SARS-CoV-2 subgenomic RNAs in routine diagnostic oropharyngeal/nasopharyngeal swabs up to 17 and 11 days after first detection by next generation sequencing (NGS) and PCR, respectively. Our finding of extended detection of subgenomic RNA in diagnostic samples has subsequently been supported by another study (available as preprint)[17] using the same E gene PCR mentioned above[16]. That very recent study detected subgenomic E RNA in swab samples from hospitalized patients up to 22 days after onset of clinical symptoms[17]. Thus, it is becoming clear that the presence, and thus detection, of SARS-CoV-2 subgenomic RNAs in diagnostic samples is rather prolonged and consequently not a good marker/indication of active virus replication/transcription or active/recent infection. Despite that, a number of high-profile studies[18–21] have continued to use presence or reduction of subgenomic RNA level as evidence of or protection from active infection, and consequently, we believe it is important to understand that these subgenomic RNAs may be present for a significant time after active infection.

## Results

**Detection and abundance of NGS reads mapped to subgenomic RNAs.** Our analysis of subgenomic RNAs included 12 SARS-CoV-2 positive swab samples and a virus-negative control sample (Table 1). Manual inspection of reads indicated the presence of subgenomic RNAs and mapping against a reference (fasta file available as Supplementary Data 1) designed to specifically map the ten potential subgenomic RNAs, indicated the presence of variable number of reads mapping to subgenomic RNAs in all SARS-CoV-2 positive samples (NCBI Sequence Read Archive (SRA): PRJNA636225) while no reads were found in the negative control sample (Table 2 and Fig. 1). Overall, of the 56 million NGS reads generated from the virus-positive samples, nearly 800,000 reads mapped to one of the ten SARS-CoV-2 subgenomic RNAs (Table 2). No reads mapped to the tentative Orf10 RNA

**Table 1 Table showing summary information about the individuals and samples included in this study.**

| Individual with gender and age group | Sample ID | NGS number | Clinical symptoms | Sample collection date | Diagnostic SARS-CoV-2 RT-PCR test (Ct Value) |
|---|---|---|---|---|---|
| 1 Female, 20–40 years | GC-28 | GC-28/67 | Fever, cough, sore throat, body pains, chest pain, non- productive cough | 28/1/20 | Not Detected |
| 2 Female, 20–40 years | GC-26 | GC-26/66 | Sore throat, dry cough | 7/3/20 | Detected (21) |
| 3 Female, 20–40 years | GC-13 | GC-13/35 | Body aches, headaches, dry cough, shortness of breath | 23/3/20 | Detected (29) |
| 4 Female, 40–60 years | GC-11 | GC-11/34 & GC-11/38 | Cold, sinusitis | 24/3/20 | Detected (19) |
| | GC-20 | GC-20/63 | Cold, sinusitis | 2/4/20 (9) | Detected (31) |
| | GC-24 | GC-24/61 | Cold, sinusitis | 7/4/20 (14) | Detected (31) |
| 5 Female, 20–40 years | GC-12 | GC-12/36 | Sore throat, rigor, fever | 24/3/20 | Detected (31) |
| 6 Female, 40–60 years | GC-14 | GC-14/33 & GC-14/37 | Unspecified | 28/3/20 | Detected (18) |
| | GC-23 | GC-23/60 | Asymptomatic | 8/4/20 (11) | Detected (31) |
| | GC-51 | GC-51/62 | Asymptomatic | 14/4/20 (17) | Detected (31) |
| 7 Female, 40–60 years | GC-21 | GC-21/64 | Shortness of breath, cough, rhinorrhoea and sore throat | 3/4/20 | Detected (31) |
| 8 Male, 40–60 years | GC-25 | GC-25/65 | Sore throat, hoarse voice | 10/4/20 | Detected (19) |
| 9 Female, 40–60 years | GC-55 | GC-55/68 | One day history of cough, no sore throat, no runny nose, no fever | 24/4/20 | Detected (16) |

Samples from a total of 9 individuals, including one control individual (Individual 1) are included. Two infected individuals (Individual 4 and 6) each had 3 samples collected at different dates. The gender and age group of each individual are shown. The days post initial sample collection are shown in brackets after the date. Sample identification and NGS sample number (barcode) are shown together with summary clinical symptoms, sampling date and results of the diagnostic SARS-CoV-2 RT-PCR test (Ct value).

and only five reads were mapped to the tentative Orf7b RNA (Table 2 and Fig. 1). In contrast, reads were mapped to the other 8 subgenomic RNAs, and although it differed among samples, S (Spike), Orf3a, and M were consistently mapped at a low level followed in increasing order by subgenomic RNAs for Orf8, Orf6, and E while Orf7a and N were mapped in the highest abundance, although this was not consistent for all samples (Table 3 and Fig. 1). The abundance, although overall more or less as expected based on assumed subgenomic RNA abundance[7–10,15], differed widely among samples, most likely depending on sample quality and overall virus genomic and subgenomic RNA abundance. Comparing samples amplified with two different polymerases (Table 3; sample GC-11/34 compared with sample GC-11/38 and GC-14/33 compared with GC-14/37) and comparing samples with longer average read length and high virus coverage (Table 3; samples GC-26/66, GC-11/38, GC-24/61, GC-14/37, and GC-23/60) did also, although with some variability from sample to sample, generate a somewhat comparable pattern. Indeed, looking at sample quality, as indicated by average read length (Table 3), strongly indicated that sample quality/read length influenced levels of subgenomic RNAs detected, likely due to these subgenomic RNA amplicons incidentally being shorter than many of the virus genome amplicons (Supplementary Table 1 and the Source Data File). To look at this, we analysed the mapping results of two samples already known to be of poor quality, having been suspended in water rather than PBS/transport medium before coming to our laboratory. Although these two samples had a high virus load in the diagnostic PCRs, the NGS generated mostly very short reads (Table 3; samples GC-25/65 and GC-55/68) and had a different pattern with a very high abundance of subgenomic RNAs dominated by the Orf7a subgenomic amplicon. This is most likely due to this amplicon being short (sequence length between leader sequence forward primer and nearest pool 2 reverse primer of only 85 nucleotides, although most other subgenomic amplicons would also be expected to be short and some genomic amplicons also being short (Supplementary Table 1 and the Source Data file). Our sample set included multiple samples from two individuals sampled 11–17 days apart and representing early and late infection (Tables 1 and 3). As can be seen when comparing those samples, subgenomic RNAs are also detected in the late infection samples and may even be preferentially amplified. Although this may possibly indicate a rather long period of virus replication/transcription, we believe it is more likely due to coronavirus membrane-associated RNAs being partly, albeit not fully, protected from host and environmental degradation (see below). Partial degradation, represented as shorter average read lengths, may result in some shorter amplicon targets being preferentially amplified (Table 2 and Supplementary Table 1 and Source Data File).

**Subgenomic RNA reads mapped to the virus genome by filtering**. To validate our results detailed above, we looked at the NGS reads to find likely subgenomic RNAs already mapped to the virus reference genome (Wuhan-Hu-1-NC_045512/MN908947.3 [https://www.ncbi.nlm.nih.gov/nuccore/NC_045512.2/ and https://www.ncbi.nlm.nih.gov/nuccore/MN908947.3]), but filtering so only reads containing part of the leader sequence were included and then looked at where these reads had been mapped. A total of between 8 and 256,123 reads containing the leader sequence were found in our positive samples while none was detected in the negative sample GC-28/67 (Supplementary Table 2). Reads were mapped to the location of the TRS of nine of the ten known subgenomic RNAs, however, only samples GC-26/66, GC-11/38, and GC-14/37 possessed reads, in a low number, mapping to the start of Orf7b. The

**Table 2 Table showing the details of the sample number, average read length, number of reads and number of reads mapped to each subgenomic SARS-CoV-2 RNA.**

| Sample | Average read length | Number of reads in millions | S Orf2 | Orf3a | E Orf4 | M Orf5 | Orf6 | Orf7a | Orf7b | Orf8 | N Orf9 | Orf10 |
|---|---|---|---|---|---|---|---|---|---|---|---|---|
| GC-28/67 | 76 | 1.8 | 0 | 0 | 0 | 0 | 0 | 0 | 0 | 0 | 0 | 0 |
| GC-26/66 | 207 | 14.3 | 1611 | 17,702 | 18,165 | 5793 | 7066 | 19,443 | 0 | 807 | 20,156 | 0 |
| GC-13/35 | 149 | 1.6 | 1 | 17 | 17 | 9 | 14 | 72 | 0 | 17 | 81 | 0 |
| GC-11/34 | 73 | 1.2 | 8 | 29 | 98 | 63 | 58 | 592 | 0 | 250 | 35 | 0 |
| GC-11/38 | 184 | 3.3 | 179 | 2464 | 1715 | 1702 | 1587 | 4107 | 0 | 406 | 5071 | 0 |
| GC-20/63 | 83 | 1.3 | 0 | 0 | 133 | 1 | 972 | 0 | 0 | 0 | 0 | 0 |
| GC-24/61 | 115 | 2.3 | 0 | 0 | 16,067 | 283 | 0 | 5132 | 0 | 0 | 14,442 | 0 |
| GC-12/36 | 163 | 1.3 | 0 | 4 | 0 | 0 | 1 | 1 | 0 | 1 | 1 | 0 |
| GC-14/33 | 153 | 6 | 15 | 57 | 77 | 99 | 60 | 261 | 0 | 42 | 174 | 0 |
| GC-14/37 | 185 | 5.2 | 243 | 6798 | 5691 | 5548 | 6551 | 12,898 | 0 | 1539 | 33,261 | 0 |
| GC-23/60 | 150 | 5.2 | 204 | 74 | 11,117 | 304 | 12,972 | 6902 | 0 | 1463 | 51,614 | 0 |
| GC-51/62 | 100 | 2.6 | 0 | 0 | 0 | 89 | 0 | 19,847 | 0 | 0 | 15,130 | 0 |
| GC-21/64 | 92 | 3 | 0 | 0 | 0 | 88 | 0 | 9408 | 0 | 1052 | 3966 | 0 |
| GC-25/65 | 84 | 5.1 | 2 | 10 | 16,581 | 1249 | 16,655 | 158,755 | 0 | 11,990 | 1841 | 0 |
| GC-55/68 | 84 | 3.9 | 13 | 21 | 35,701 | 4453 | 4265 | 166,578 | 5 | 3328 | 942 | 0 |

**Fig. 1 SARS-CoV-2 genomic and subgenomic RNA structure showing genes and open reading frames (ORF) together with violin plots showing the number of reads per total of 5 million reads in the diagnostic samples mapped to the leader-containing subgenomic RNAs in the fasta file used for mapping.** The median read count is indicated by the white dot, the interquartile range (IQR) by the thick bar, and the furthest values within 1.5*IQR indicated by the thin black line ($n = 14$ data points from 12 biological samples from eight individuals run once). The structure of the SARS-CoV-2 genomic RNA is shown at the top and each subgenomic RNA is illustrated next to the respective violin plot. The nucleotide positions for the leader-TRS (transcription-regulatory sequences) joining locations are indicated alongside each subgenomic RNA (numbering based on reference Wuhan-Hu-1 NC_045512.2/MN908947.3 [https://www.ncbi.nlm.nih.gov/nuccore/NC_045512.2/ and https://www.ncbi.nlm.nih.gov/nuccore/MN908947.3]). In the current study no reads mapped to the tentative Orf10 leader-containing subgenomic RNA.

number of reads with a leader sequence mapped to the corresponding ORF in the SARS-CoV-2 genome are shown in Supplementary Table 2 and Supplementary Fig. 1. While, the percentages varied among the samples, the three subgenomic RNAs with the highest median number of reads with the leader sequence were the E gene/Orf4 (4.1%), Orf7a (17.4%), Orf8 (4.3%) and N gene/Orf9 (10.7%) (Supplementary Fig. 1).

The samples with the highest number of reads mapping to cryptic or unknown TRS were the poorer quality samples GC-11/34, GC-21/64, and GC-25/65 and no consistent pattern was observed in the mapping of reads with the leader sequence to any individual unrecognized TRS site.

**Searching the NCBI SRA for reads mapping to subgenomic RNAs.** Another step in our analysis included searching the NCBI SRA and selection of a few deposited NGS reads from studies

using either the same SARS-CoV-2 Ampliseq panel or generated by other methods. Although not abundant for all of them, reads representing subgenomic RNAs rather than virus genomic RNA could be found by simple analysis using e.g., BlastN. Again, as in our own data, we detected no or very little subgenomic RNA of Orf7b and no evidence for Orf10 subgenomic RNA.

To look at this in more detail, we downloaded a selection of SRA's generated from different sample types, different sequencing platforms and employing different library strategies. Reads belonging to subgenomic RNA could be identified in all samples except sample (SRR11454612) from RNAseq on a sputum sample from an infected human (Supplementary Table 3). The two selected Ion Torrent Ampliseq SRA's (SRR11810731 and SRR11810737) produced the highest number of subgenomic reads, followed by an RNA-Seq experiment performed in cell culture using a Nanopore platform (SRR11267570). The selected RNA-Seq experiments performed on clinical samples typically

**Table 3 Table showing the details of the sample number, average read length, number of reads and number of reads mapped to each subgenomic SARS-CoV-2 RNA adjusted so they represent reads normalized to a total of 5 million (5 M) reads for each sample for easier comparison.**

| Sample | Average read length | Number of reads adjusted to 5 M reads | S Orf2 | Orf3a | E Orf4 | M Orf5 | Orf6 | Orf7a | Orf7b | Orf8 | N Orf9 | Orf10 |
|---|---|---|---|---|---|---|---|---|---|---|---|---|
| GC-28/67 | 76 | 5 M | 0 | 0 | 0 | 0 | 0 | 0 | 0 | 0 | 0 | 0 |
| GC-26/66 | 207 | 5 M | 563 | 6190 | 6351 | 2026 | 2471 | 6798 | 0 | 282 | 7048 | 0 |
| GC-13/35 | 149 | 5 M | 3 | 53 | 53 | 28 | 44 | 225 | 0 | 53 | 253 | 0 |
| GC-11/34 | 73 | 5 M | 33 | 121 | 408 | 263 | 242 | 2467 | 0 | 1042 | 146 | 0 |
| GC-11/38 | 184 | 5 M | 271 | 3733 | 2598 | 2579 | 2405 | 6223 | 0 | 615 | 7683 | 0 |
| GC-20/63 | 83 | 5 M | 0 | 0 | 512 | 4 | 3738 | 0 | 0 | 0 | 0 | 0 |
| GC-24/61 | 115 | 5 M | 0 | 0 | 34,928 | 615 | 0 | 11,157 | 0 | 0 | 31,396 | 0 |
| GC-12/36 | 163 | 5 M | 0 | 15 | 0 | 0 | 4 | 4 | 0 | 4 | 4 | 0 |
| GC-14/33 | 153 | 5 M | 13 | 48 | 64 | 83 | 50 | 218 | 0 | 35 | 145 | 0 |
| GC-14/37 | 185 | 5 M | 234 | 6537 | 5472 | 5335 | 6299 | 12,402 | 0 | 1480 | 31,982 | 0 |
| GC-23/60 | 150 | 5 M | 196 | 71 | 10,689 | 292 | 12,473 | 6637 | 0 | 1407 | 49,629 | 0 |
| GC-51/62 | 100 | 5 M | 0 | 0 | 0 | 171 | 0 | 38,167 | 0 | 0 | 29,096 | 0 |
| GC-21/64 | 92 | 5 M | 0 | 0 | 0 | 147 | 0 | 15,680 | 0 | 1753 | 6610 | 0 |
| GC-25/65 | 84 | 5 M | 2 | 10 | 16,256 | 1225 | 16,328 | 155,642 | 0 | 11,755 | 1805 | 0 |
| GC-55/68 | 84 | 5 M | 17 | 27 | 45,771 | 5709 | 5468 | 213,562 | 6 | 4267 | 1208 | 0 |

generated very low levels of reads mapping to the virus genome and consequently to the leader sequence. The Artic network primers[22] also detected subgenomic reads in virus culture experiments (ERR4157962 and ERR4157960).

The subgenomic RNAs with the highest number of reads mapped in the SRA's were the N and Orf7a followed by the Orf3a and M gene. The subgenomic S gene and Orf6 were typically low and no reads were mapped to the subgenomic Orf10 in any sample. Only sample SRR11267570 and SRR11810737 had any reads mapped to the subgenomic Orf7b (0.2–0.3% of reads having the leader sequence).

**Further abundance analysis of mapped NGS amplicons.** The number of reads mapped to either the first 21,500 nucleotides (nt) of the reference virus genome, to the subgenomic region from nucleotide 21,500 onward, to subgenomic RNA containing the leader sequence, to the included cellular control mRNA amplicons and reads not mapped to any of these are summarized in Fig. 2. Specific details about the abundance of cellular mRNA amplicons in each NGS sample are shown in Table 4. Some samples have very few reads mapped to cellular mRNA amplicons, e.g., samples GC-25/65 and GC-55/68 having been submitted in water, while other samples, such as the low virus load samples GC-23/60, GC-24/61, GC-51/62, GC-20/63, and GC-21/64 and the negative control sample GC-28/67, have many reads mapped to cellular mRNA amplicons (Table 4 and Fig. 2). Interestingly, samples GC-14/33/37 and GC-11/34/38 also had a low number of reads mapped to cellular mRNA amplicons. These samples have a high SARS-CoV-2 load and were taken early in infection and this may also be the case for sample GC-26/66 (Table 4), consistent with a likely reduced level of cellular mRNAs in early, high virus load infection (Table 4 and Fig. 2).

We then looked further at the range of reads mapped to SARS-CoV-2 amplicons in the samples, as the abundance of different size amplicons could possibly be influenced by sample quality or virus load, in particular in poor quality or low virus load samples. The results are shown in the last 3 columns of Table 4 (and further details in the Source Data file). From that data it is evident that certain amplicons are highly abundant, and those amplicons are consistently short amplicons of 68–78 nucleotides (excluding primers) located in the Orf1ab region of the virus genome (the number given in the third last column in Table 4), and thus amplified from genomic RNA, or from one of a few relatively short or from a very short amplicon of 54 nucleotides located in the 3′-end of the virus genome (the number given in the second last column in Table 4), and thus amplifying both genomic and

subgenomic RNAs. The average coverage for the 168 amplicons included in the first 21,500 nt of the virus genome is shown in the last column of Table 4.

**Abundance of reads mapped to virus genomic or subgenomic amplicons.** Due to the variability in individual amplicon abundances observed, we then compared the abundance of reads mapped to SARS-CoV-2 genomic amplicons to the abundance of reads mapped specifically to subgenomic RNAs. This comparison was done in a number of different ways including a comparison of the total reads for the first 21,500 nucleotides of the virus genome with the total number of subgenomic reads; the most abundant amplicon in the first 21,500 nt of the virus genome with the most abundant subgenomic RNA amplicon reads; the most abundant full virus genome amplicon with the most abundant subgenomic RNA amplicon reads; and the average full virus amplicons reads with the average subgenomic RNA amplicons reads as presented in Fig. 3 (further details in Supplementary Figs. 2 and 3 and in the Source Data file). Although the ratio was variable and differed depending on the way of comparison, the overall median ratio was in the 2.3–24.3 range, somewhat similar to what was estimated using PCR, see below.

**SARS-CoV-2 PCR assays to detect specific targets.** The results for PCR testing of samples for detection of specific genomic and subgenomic targets are shown in Table 5 and Fig. 4. Of the 12 initial diagnostic positive samples available for testing, 11 were still positive while a single sample previously tested weak positive by PCR and having some SARS-CoV-2 reads by NGS, sample GC-12/36, was now negative, consistent with that sample initially being borderline positive and the cDNA further diluted for this additional PCR testing likely lowering sensitivity (Table 5 and Fig. 4). Of the five samples negative in the 7a subgenomic PCR, this corresponded to the NGS reads for the 7a subgenomic RNA in two of these samples being low or zero (4 and 0 reads per 5 mill NGS reads for samples GC-12/36 and GC-20/63, respectively (Table 3)). However, the three other samples being negative in this PCR (Table 5), samples GC-24/61, GC-51/62, and GC-21/64, had more than 10,000 reads per 5 million NGS reads mapped to the 7a subgenomic RNA by NGS (Table 3), indicating that the NGS method is more sensitive than PCR for this purpose. This is consistent with these samples only being borderline positive in the other PCRs (Table 5). Interestingly, one sample, sample GC-13/35, that had relatively few 7a subgenomic reads detected by NGS (225 reads per 5 million NGS reads; Table 3) was weak positive by the 7a subgenomic PCR (Table 5). Overall, the 7a subgenomic PCR was only able to detect the target up to 11 days

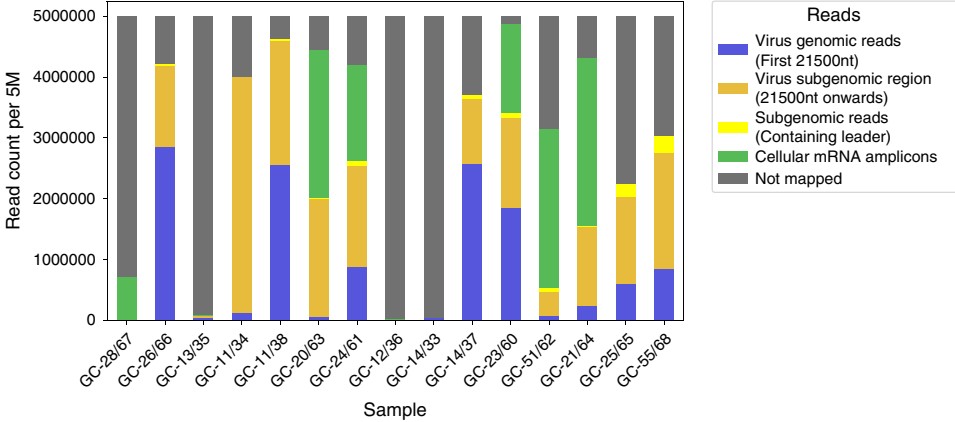

**Fig. 2 Read count per 5 million (5 M) showing reads mapped to either the first 21,500 nucleotides (nt) of the virus genome, to the subgenomic region from nucleotide 21,500 onward, to subgenomic RNA containing the leader sequence, to the included cellular control mRNA amplicons and reads not mapped to any of these.** $n = 15$ data points from 13 biological samples from nine individuals run once.

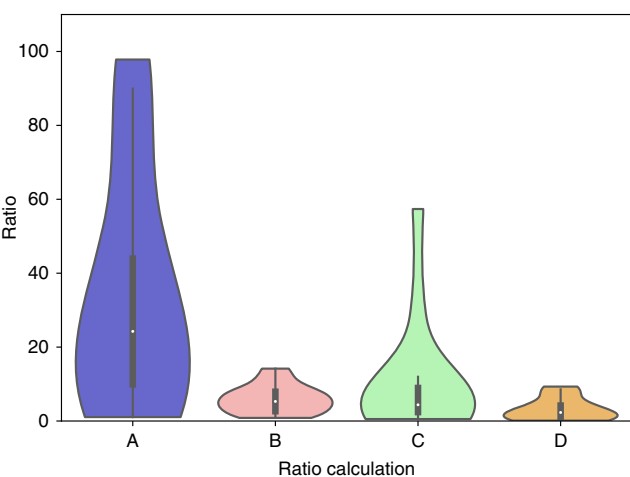

**Fig. 3 Violin plot showing the estimated ratio of virus genomic reads to subgenomic reads containing the leader for each of the diagnostic samples included in the study.** The ratio is estimated in different ways, including comparing **a** the total reads for the first 21,500 nucleotides of the virus genome with the total number of subgenomic reads; **b** the most abundant amplicon in the first 21,500 nt of the virus genome with the most abundant subgenomic RNA amplicon reads; **c** the most abundant full virus genome amplicon with the most abundant subgenomic RNA amplicon reads; and **d** the average full virus amplicons reads with the average subgenomic RNA amplicons reads. The median read count ratio is indicated by the white dot, the interquartile range (IQR) by the thick bar, and the furthest values within 1.5*IQR indicated by the thin black line ($n = 14$ data points from 12 biological samples from eight individuals run once).

after first detection while the NGS method also detected a sample taken 17 days after first detection, the last time point included in our study. It should be mentioned, that we had to dilute the cDNA used for these PCRs as we had limited amounts available.

The differences in Ct values between the 7a subgenomic and the genomic targets for the samples with a positive PCR is shown in Table 5 and Supplementary Fig. 4. With the amplification efficiency of these assays the difference corresponds to around 4-fold to 20-fold more of the genomic target than of the 7a subgenomic RNA-specific PCR target.

**Strand specific PCR**. For strand specific PCR, we focused on the seven samples that were most likely to have a sufficient virus RNA load to allow potential detection of the strand-specificity of the detected SARS-CoV-2 RNA. While all seven samples were positive for positive sense SARS-CoV-2 RNA, sample GC-23/60 sampled 11 days after first detection (sample GC-14/33/37) was only borderline positive, and only for positive sense subgenomic RNA close to the level of detection of these assays (sensitivity approximately 10-fold lower than the non-strand specific PCRs). Positive sense genomic RNA was only detected at a relatively low level in the PCR on sample GC-13/35 (Supplementary Table 4). The identity of positive and negative sense amplicons obtained for sample GC-14/33/37, GC-25/65, and GC-55/68 were confirmed by Sanger sequencing. Only three samples were weak positive for negative sense SARS-CoV-2 genomic RNA and of these, only two had a borderline signal for the negative sense 7a subgenomic target. However, it is worth mentioning that these three samples included one known to have been taken early in infection, i.e., sample GC-14/33/37, while the other two were samples with a very high virus load according to all diagnostic PCRs and submitted in water rather than in PBS or transport fluid as for the other samples (Samples GC-25/65 and GC-55/68, Supplementary Table 4).

For the samples with a sufficient load for detection by this method, the difference in Ct values for plus strand detection between the 7a subgenomic and genomic targets was slightly higher than for the nonstrand specific PCRs mentioned in the section above, and for the amplification efficiency of these assays corresponds to a difference of around 14-fold to 28-fold more of the plus strand genomic compared to the 7a subgenomic RNA-specific PCR target. Although this difference in detection of either sense (plus or negative sense) or only positive sense of the subgenomic 7a RNA is small, it may possibly indicate that more of the subgenomic RNA as compared to the genomic RNA is of negative sense. Although the number of samples is very low, this is supported by the fact that the difference in Cts obtained between the positive sense and negative sense genomic target is 8.6–8.8 Cts (150-fold more positive sense RNA), while it is only five Cts (20-fold more positive sense RNA) for the subgenomic RNA in the two samples for which detectable levels were present ((Samples GC-25/65 and GC-55/68), Supplementary Table 4). Furthermore, comparing the Ct values of negative strand genomic to negative strand subgenomic for these two samples, the difference is only 0.9–1.9 Ct, consistent with only around 2-fold

**Table 4 Table showing the number of NGS reads, per 5 mill reads, mapped to control cellular mRNA amplicons and the highest and average number of reads for SARS-CoV-2 amplicons included in the Ampliseq panel.**

| Sample | Sample collection date | SARS-CoV-2 PCR (Ct) | TBP | LRP1 | HMBS | MYC | ITGB7 | Total | Mill reads | Total per 5 M reads | Virus amplicon coverage per 5 mill reads | | |
|---|---|---|---|---|---|---|---|---|---|---|---|---|---|
| | | | | | | | | | | | Highest in first 21,500 nt | Highest in full virus genome | Average of 168 amplicons in first 21,500 nt |
| GC-28/67 | 28/1/20 | Not detected | 157,551 | 2944 | 70,221 | 22,633 | 16 | 253,365 | 1.8 | 703,792 | 0 | 0 | 0 |
| GC-26/66 | 7/3/20 | Detected (21) | 1486 | 193 | 476 | 338 | 732 | 3225 | 14.3 | 1128 | 99,927 | 116,186 | 48,581 |
| GC13/35 | 23/3/20 | Detected (29) | 54 | 2 | 22 | 12 | 120 | 210 | 1.6 | 656 | 3022 | 26,200 | 185 |
| GC-11/34 | 24/3/20 | Detected (19) | 0 | 0 | 0 | 1 | 1 | 2 | 1.2 | 8 | 12,617 | 3,776,575 | 726 |
| GC-11/38 | 24/3/20 | Detected (19) | 0 | 9 | 0 | 0 | 2 | 11 | 3.3 | 17 | 55,377 | 373,552 | 15,201 |
| GC-20/63 | 2/4/20 | Detected (31) | 259,450 | 11,510 | 35,621 | 43,798 | 284,859 | 635,238 | 1.3 | 2,443,223 | 28,396 | 1,337,769 | 72 |
| GC-24/61 | 7/4/20 | Detected (31) | 222,864 | 82,467 | 66,551 | 138,145 | 220,593 | 730,620 | 2.3 | 1,588,304 | 181,672 | 469,278 | 9510 |
| GC-12/36 | 24/3/20 | Detected (31) | 421 | 66 | 28 | 626 | 1725 | 2866 | 1.3 | 11,023 | 81 | 900 | 4 |
| GC-14/33 | 28/3/20 | Detected (18) | 0 | 1 | 0 | 0 | 1 | 2 | 6 | 2 | 693 | 1880 | 172 |
| GC-14/37 | 28/3/20 | Detected (18) | 1 | 18 | 0 | 2 | 3 | 25 | 5.3 | 24 | 79,092 | 91,491 | 15,358 |
| GC-23/60 | 8/4/20 | Detected (31) | 214,567 | 81,749 | 91,585 | 163,657 | 214,265 | 765,823 | 2.6 | 1,472,737 | 408,398 | 661,887 | 21,965 |
| GC-51/62 | 14/4/20 | Detected (31) | 483,313 | 260,623 | 210,787 | 159,479 | 242,317 | 1,356,519 | 2.6 | 2,608,690 | 32,142 | 269,108 | 222 |
| GC-21/64 | 3/4/20 | Detected (31) | 575,261 | 167,868 | 230,305 | 96,124 | 580,411 | 1,649,969 | 3 | 2,749,948 | 125,907 | 704,475 | 827 |
| GC-25/65 | 10/4/20 | Detected (19) | 5 | 7 | 85 | 186 | 1055 | 1338 | 5.1 | 1312 | 351,893 | 1,129,815 | 3614 |
| GC-55/68 | 24/4/20 | Detected (16) | 2 | 0 | 0 | 0 | 0 | 2 | 3.9 | 3 | 417,778 | 946,012 | 3922 |

more negative strand genomic than subgenomic RNA (Supplementary Table 4). This estimate is based on only two samples, both of which have a very low Ct (high virus target load) in the diagnostic PCRs and both inadvertently having been submitted in water.

To analyse if samples potentially contained SARS-CoV-2 double stranded RNA, we attempted treatment of samples with the single-stranded RNase If before strand-specific PCRs. The selected samples included samples GC-26/66, GC-11/34/38, GC-14/33/37, GC-25/65, and GC-55/68 and the two strongest fractions from the membrane association/fractionation resistance protocol described below. However, after treatment with RNase If, which is described to have a preference for degradation of single-stranded RNA over double-stranded RNA, these samples were completely negative for both positive and negative strand SARS-CoV-2 RNA by PCR. This may be due to a number of factors, including double stranded RNA being below detection limits or possibly that virus plus and negative strands may not have properly annealed before the RNase treatment. Alternatively, the relatively high RNase If concentration used or other nucleases present during incubation of extracted nucleic acids samples at room temperature and at 37 °C in nuclease buffer may have destroyed any double stranded RNA present. We are not able to further look into this as our sample material is now exhausted. However, further studies could look at this in infected cell cultures.

**Membrane association and nuclease resistance of SARS-CoV-2 RNAs.** The two samples selected for this analysis represented two different types of samples. Sample GC-26/66 representing a good quality sample and sample GC-55/68, having been suspended in water rather than in PBS or transport medium, representing a sample with partly degraded RNA.

As part of this protocol, half of the final 16 fractions obtained for each sample were treated with Triton X-100 to determine whether lipid membranes may be important in protecting any SARS-CoV-2 RNA present. As hypothesized, Triton X-100 treatment had a significant effect on final fractions. The 16 treated fractions became either negative or only borderline positive for all three targets in the commercial PCR used, i.e., with Cts in the mid to upper thirties very close to the detection limit of the PCR (see the Source Data file for details). Overall, this indicates that the Triton X-100 treatment, even without addition of external nucleases, results in degradation of any SARS-CoV-2 RNA present by at least 1000-fold or more, consistent with such RNA being protected by lipid membranes and consistent with what has been observed for SARS-CoV replication/transcription complexes in cell culture[23].

Of the non-Triton X-100 treated fractions, 15 out of 16 had detectable levels of SARS-CoV-2 RNA for all 5 PCR targets used. The only fraction with a single negative result for the 7a-subgenomic PCR was sample GC-26/66 S1S10T-N+ (supernatant from initial 1000 × g spin, non-Triton treated, nuclease treated and then supernatant from final 10,000 × g spin). This particular fraction was positive for the other PCR targets, see Supplementary Table 5. Interestingly, the good quality sample GC-26/66, ended up with most of the SARS-CoV-2 targets in the final pelleted fractions and these targets being highly resistant to nuclease treatment (Supplementary Table 5 and Supplementary Fig. 5). In fact, the nuclease treated fraction (GC-26/66 P1P10T-N+) had a lower Ct, i.e., higher target load than the non-nuclease treated fraction, a phenomenon we have observed earlier for highly purified, nuclease resistant targets[24]. In contrast, sample GC55/68 that had been in water rather than PBS/transport medium, had most of the targets in the supernatant fraction from the first 1000 × g spin, and what was

| Sample | Sample collection date | Leader-7a sub-genomic (Set 1) (Ct) | 7a Genomic and sub-genomic (Set 2) (Ct) | Leader-5′-UTR genomic (Set 3) (Ct) | 5′-UTR genomic (Set 4) (Ct) | N Gene (Ct) | Orf 1ab (Ct) | S gene (Ct) |
|---|---|---|---|---|---|---|---|---|
| NTC | | Neg | Neg | Neg | Neg | Neg | Neg | Neg |
| GC-28/67 | 28/01/2020 | Neg | Neg | Neg | Neg | Neg | Neg | Neg |
| GC-26/66 | 7/03/2020 | 28.9 | 24.8 | 25.7 | 25.9 | 25.7 | 26.2 | 26.7 |
| GC-13/35 | 23/03/2020 | 32.2 | 28.6 | 29.6 | 28.8 | 29.6 | 28.0 | 29.1 |
| GC-11/ 34/38 | 24/03/2020 | 24.3 | 19.7 | 20.6 | 21.4 | 21.3 | 21.9 | 22.4 |
| GC-20/63 | 2/04/2020 | Neg | 35.3 | 35.6 | 35.7 | 35.8 | Neg | Neg |
| GC-24/61 | 7/04/2020 | Neg | 34.8 | 33.8 | 36 | 35.4 | 35.3 | Neg |
| GC-12/36 | 24/03/2020 | Neg | Neg | Neg | Neg | Neg | Neg | Neg |
| GC-14/ 33/37 | 28/03/2020 | 21.1 | 17.5 | 17.8 | 18.6 | 19.3 | 19 | 19.7 |
| GC-23/60 | 8/04/2020 | 35.9 | 33.3 | 32.9 | 33.3 | 33.6 | 32.9 | 35.2 |
| GC-51/62 | 14/04/2020 | Neg | Neg | Neg | 35.9 | 37 | Neg | Neg |
| GC-21/64 | 3/04/2020 | Neg | Neg | Neg | 36.8 | 36.3 | Neg | Neg |
| GC-25/65 | 10/04/2020 | 22.3 | 18.9 | 20.1 | 21 | 20.4 | 21.7 | 22.2 |
| GC-55/68 | 24/04/2020 | 19.3 | 15 | 15.7 | 16.1 | 16.4 | 16.7 | 17.5 |

**Table 5 Table showing sample details with corresponding Ct values of PCR amplification using specific targets.**

*NTC* non-template control (water).

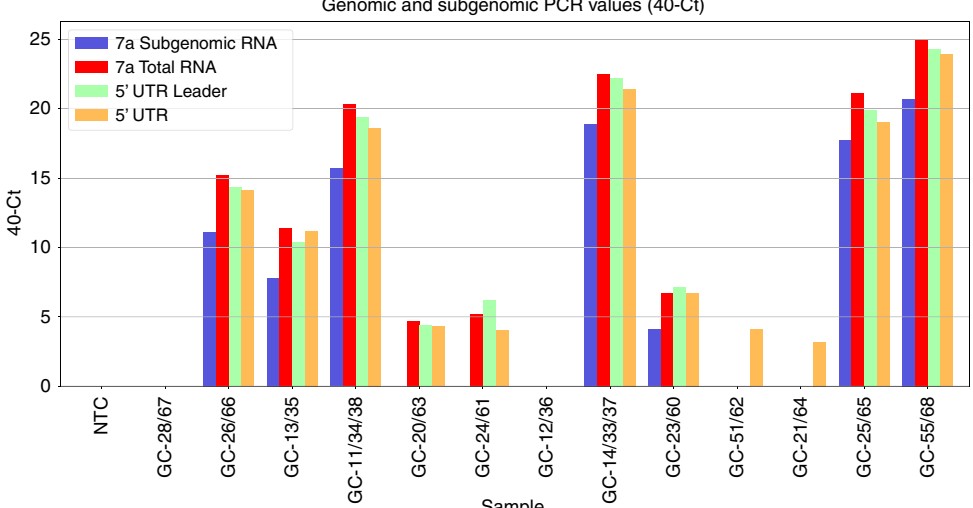

**Fig. 4 Genomic and subgenomic PCR values shown as 40 minus Ct (40-Ct).** The values are shown for each sample and for the four in-house PCRs detecting 7a subgenomic RNA only, 7a total RNA (i.e., genomic RNA and subgenomic RNA up to and including 7a), the 5′-UTR (untranslated region) leader or the 5′UTR. NTC non-template control (water).

present in this supernatant was partly susceptible to nuclease treatment and could not be efficiently pelleted by the 10,000 × *g* spin. Furthermore, for this sample the target RNA present in the initial 1000 × *g* pellet was, in contrast to what was observed for sample GC-26/66, highly susceptible to nuclease treatment (Supplementary Table 5 and Supplementary Fig. 5).

Looking at the detected levels of PCR targets for the 7a subgenomic RNA target compared to the 7a genomic target (that detect both genomic and subgenomic targets), the difference was somewhat similar to what is shown in the section above for results directly on samples and corresponding to having around ten times more genomic than subgenomic targets (Table 5 and Supplementary Table 5). An exception to this pattern may be the initial supernatants subjected to nuclease treatment and then pelleted at 10,000 × *g* (S1P10T-N+). For sample GC-26/66, this fraction had a difference of 5.5 Cts and for sample GC-55/68, a

difference of 11 Cts possibly indicating a higher proportion of nuclease protected virion RNA as compared to subgenomic RNAs in this particular fraction (Supplementary Table 5). This would be consistent with the strand specific PCR results for sample GC-55/ 68 mentioned above, where the Ct difference between positive strand genomic and subgenomic RNA was 8.8 Cts (Supplementary Table 4), and thus also consistent with a high proportion of positive sense virion RNA in that sample.

To look at this in more details, we also did the strand-specific PCR for these fractions. However, the sensitivity of the strand-specific assay (approximately 10-fold less sensitive than the non-strand specific assay) on fractionated samples was not sufficient to detect any negative sense SARS-CoV-2 RNA. However, we could detect positive sense genomic plus subgenomic RNA in most of the fractions in an amount consistent with expected levels detected in the non-strand specific PCRs (Supplementary Table 5).

## Discussion

We here describe the specific detection and mapping of SARS-CoV-2 leader-containing subgenomic RNAs in routine diagnostic oropharyngeal/nasopharyngeal swabs subjected to next generation sequencing (NGS). We present results from two different approaches to map subgenomic RNAs, one mapping directly to the expected sequences of the leader containing subgenomic RNAs and another approach, where reads already mapped to the virus reference genome are filtered based on whether they contain the partial leader sequence or not. We also analyse for the presence of subgenomic RNAs in selected read archives from the NCBI Sequence Read Archive. Furthermore, we extend our study to include semiquantitation of subgenomic and genomic SARS-CoV-2 RNAs by specific PCRs and quantitation of plus strand as compared to negative strand SARS-CoV-2 RNAs. Finally, we present results supporting that these subgenomic RNAs are associated with cellular membranes and are nuclease resistant. Aided by the current understanding of the cell biology of coronavirus infections including the known association of virus RNAs with cellular double-membrane vesicles (DMVs), we present an integrated interpretation of our results based on detailed analysis of relative abundance of the different subgenomic RNAs in samples collected early and late in infection, of different quality or subjected to membrane lysis by detergent treatment and fractionation. Our integrated interpretation of the results overall, is that both virion and subgenomic RNAs are most likely rather stable in vivo and that detection of subgenomic RNAs in clinical samples, importantly, do not necessarily signify active virus replication/transcription, but instead is due to such RNAs being part of membrane vesicles, most likely so-called double-membrane vesicles, and thus relatively stable.

Our mapping of specific subgenomic RNAs indicated that samples had variable levels of eight of the predicted ten subgenomic RNAs while no or very low levels of subgenomic RNA for Orf7b was detected and a subgenomic RNA for Orf10 was absent. This is consistent with what has been described for SARS-CoV-2 in cell culture[9]. While some samples, in particular those with a high virus load, had very few reads mapped to cellular RNA control amplicons, other samples, in particular the negative sample and those with a low virus load, had many reads mapped to these amplicons. When comparing the number of reads from SARS-CoV-2 genomic amplicons to the number of reads mapped specifically to the SARS-CoV-2 subgenomic RNAs, we found that the ratio of genomic to subgenomic RNA reads varied from around 2–20 depending on the specific comparison made. This finding was consistent with the results from PCR with specific targets, which indicated a ratio of genomic to 7a subgenomic RNA of around 4–20 fold. Further testing of these samples using so-called strand-specific PCRs indicated that for the positive sense RNA, the ratio of the 7a genomic to subgenomic RNA is around 14–28 fold while the positive to negative sense ratio for the 7a subgenomic RNA is around 20-fold and around 150-fold or higher for the genomic RNA. Although the presence of both negative and positive sense RNA in some samples indicated that double-stranded forms of these RNAs may be present, the limited sample volumes available and the lower sensitivity of these methods did not allow us to detect that.

Another aspect evaluated in this study, was whether the SARS-CoV-2 RNAs detected in the diagnostic samples were protected from nucleases, and whether such protection was likely to be facilitated by cellular membranes as hypothesized. The protocol for this part of the study was based on a study of SARS-CoV replication/transcription complexes in cell culture that showed that such membrane complexes could be pelleted by centrifugation and protected the virus RNA from nucleases unless disrupted by mild detergent treatment[23]. With a slight modification of this protocol, we were able to show, similar to the original cell culture study of SARS-CoV, that the SARS-CoV-2 RNAs were, at least in part, protected from nucleases, could be pelleted by centrifugation and that detergent treatment would greatly reduce the nuclease protection and ability to be pelleted by centrifugation. Interestingly, even after fractionation, the ratio of genomic to subgenomic 7a RNA was still around 10-fold except for a fraction thought to mainly include nuclease protected virion RNA, for which the ratio may be as high as 600-fold more genomic to subgenomic RNA.

The results described here fully support that SARS-CoV-2 genomic and subgenomic RNAs are present in diagnostic samples even in late infection/after active infection. Subgenomic RNAs, like virion RNA, are rather stable and are likely protected from nucleases by cellular membranes, for the subgenomic RNAs possibly the so-called double-membrane vesicles known to support coronavirus RNA replication and transcription[7,8,23,25]. Detection of subgenomic RNAs in late infection, as described here up to 11 and 17 days after first detection by PCR and NGS, respectively, although in contrast to the studies by Wölfel et al.[16], is consistent with the recent findings described by van Kampen et al.[17], which detected the E gene subgenomic RNA by PCR in respiratory swabs up to 22 days after first day of onset of clinical symptoms. The participants in their study likely had more severe disease than the ones included in our study, as their study focused on hospitalized patients, many of which were in intensive care units, while our study subjects only had minor clinical symptoms and all self-isolated at home[26]. Nevertheless, although their study detected the E gene subgenomic RNA by PCR while we focused on the 7a subgenomic RNA by PCR and all the subgenomic RNAs by NGS, these studies support each other and were also supported by results presented in additional studies using the same subgenomic E gene PCR[18–21]. The detection of subgenomic RNA is therefore not direct evidence of active infection, instead its presence at lower levels than virion genomic RNA results in detection for a shorter period of time unless using e.g., highly sensitive NGS.

Overall, our results fit well with what would be expected from a coronavirus infection based on what is known from cell culture studies. The caveat is that samples from even relatively early infection in vivo, as assessed by upper respiratory swab samples, are more alike late infection cell culture supernatant or partly purified virion preparations and less like what is found for early intracellular coronavirus RNAs in cell culture. Consequently, when looking at what is known for other coronaviruses and cell culture studies, intracellular subgenomic RNAs may dominate over genomic RNA very early on, with 8–70 times more intracellular subgenomic than genomic RNA at 6–8 h after infection for infectious bronchitis virus, a gammacoronavirus, and for bovine coronavirus, a betacoronavirus, and with at least ten times more plus sense than minus sense RNA[12,15,25]. In contrast, the same authors found that extracellular and partly purified coronavirus virion preparations from late cell culture infection, while being RNase resistant and susceptible to detergents, have a much higher genomic to subgenomic RNA ratio of 10–30 or higher and at least 100-fold more positive than negative sense RNA[12,15,25]. Consequently, our findings based on NGS, specific PCR assays and fractionation together with nuclease and detergent treatment, are fully consistent with what has been shown from cell culture infection and fractionation of coronavirus replication/transcription complexes in cellular membrane structures, most likely double-membrane vesicles (DMVs). Thus, SARS-CoV-2 RNAs in diagnostic swab samples are likely found as a mixture of virion genomic as well as subgenomic RNAs, both protected from nucleases by virus/cellular membranes and at a ratio of around 10-fold more genomic/virion RNA than subgenomic RNA and a

plus to minus sense RNA ratio of around 150-fold or more for genomic/virion RNA and around 20 for subgenomic RNA. This stability of subgenomic RNAs together with the variability observed for different amplicons at low target levels, may at least in part help explain variability/discrepancies of PCR results reported for different diagnostic PCR assays detecting targets in different parts of the SARS-CoV-2 genome[27,28]. For example, our analysis indicated that some subgenomic RNAs may be more abundantly amplified in poor samples, possibly because of partly degraded RNA in such samples and the increased ability of PCR, including most diagnostic PCRs as well as NGS employing various amplification steps, to amplify short targets. This notion is supported by our findings in two diagnostic samples that for unrelated reasons had been suspended in water rather than in PBS or transport medium as for our other samples. These particular two samples (sample GC-25/65 and GC-55/68), in which cells and membrane vesicles were almost certainly partly disrupted[29], and thus any coronavirus RNAs exposed to the environment and likely to RNases, were dominated by short reads, i.e., short amplicons, and strikingly, with a very high abundance of reads mapped to some subgenomic RNA amplicons, in particular the Orf7a RNA. Clearly, original sample abundance cannot change just because cellular membranes are lysed by the hypotonic treatment, so the observed increase in these reads may almost certainly be caused by preferred amplification of shorter or more efficient amplicons in such samples.

In conclusion, by combining knowledge of general coronavirus cell biology and replication/transcription with careful mapping of NGS reads to SARS-CoV-2 subgenomic RNAs and by PCR on clinical samples taken at different times of infection and of different quality, we present information that helps understand prolonged and sometimes inconsistent PCR-positivity. This information may in turn pave the way for development of better diagnostic PCRs and NGS strategies to define active SARS-CoV-2 infection as opposed to extended presence of what most likely represent highly stable virus genomic and subgenomic RNAs present in, and at least in part protected by, cellular membranes; for the subgenomic RNAs most likely so-called double-membrane vesicles (DMVs). Our findings are likely to be relevant also for other coronaviruses and possibly also other viruses in the Order *Nidovirales*. That coronaviruses, and their RNA, may be extremely resistant when part of a membrane matrix is well known, and was demonstrated for example when porcine epidemic diarrhea virus, also a coronavirus, entered and infected pigs in Canada in early 2014 by feed containing spray-dried porcine plasma[30,31]. Detergent treatment and ultracentrifugation indicated that this coronavirus RNA was initially bound to membranes, but could be pelleted by ultracentrifugation after detergent release. Other studies, including our own using yet another coronavirus, the avian infectious bronchitis virus, further support that detergent treatment will release the coronavirus RNA and make it susceptible to nuclease degradation supporting the fact that the majority of such coronavirus RNA is membrane bound[24]. Consequently, we believe that the methods described here to detect and look at relative abundance of SARS-CoV-2 RNAs in clinical samples together with insights in what is known about coronavirus cell biology overall, will have general interest and applicability not only for SARS-CoV-2, but also for other coronaviruses and related viruses.

## Methods

**Samples**. We here describe detailed analysis of samples subjected to next generation sequencing (NGS) at the Geelong Center for Emerging Infectious Diseases (GCEID). The study included combined nasopharyngeal and oropharyngeal swab samples collected from individuals in the region of Greater Geelong, Victoria, Australia between the 28th of January to the 24th of April 2020. The study included

NGS of 12 SARS-CoV-2 PCR-positive samples and 1 negative sample as control. The 12 PCR-positive samples were obtained from eight individuals as NGS was done on samples taken at three time points from two individuals to monitor their infection[26]. Summary details of the samples included are shown in Table 1. The study complied with all relevant ethical regulations and has been approved by the Barwon Health Human Research Ethics Committee (Ref HREC 20/56) and all participants gave their informed consent.

In addition to analysis of the NGS reads obtained from the samples mentioned above, we also searched the National Center for Biotechnology Information (NCBI) Sequence Read Archive (SRA [https://www.ncbi.nlm.nih.gov/sra/]) and used selected SRA studies to support the findings from our own samples.

**Nucleic acid extraction, cDNA synthesis, and SARS-CoV-2 Ampliseq NGS**. Nucleic acid extraction and cDNA synthesis was performed by heating extracted nucleic acids at 70 °C for 5 min and rapid cooling on ice before cDNA synthesis using SuperScript™ VILO™ Master Mix (Thermofisher Scientific, Victoria, Australia) as per manufacturers' instructions[26,32]. Prepared cDNA samples were then amplified using the Ion Ampliseq™ Library Kit 2.0 (Thermofisher Scientific, Victoria, Australia)[33,34] and a commercially available SARS-CoV-2 Ampliseq panel [https://www.thermofisher.com/au/en/home/life-science/sequencing/dna-sequencing/microbial-sequencing/microbial-identification-ion-torrent-next-generation-sequencing/viral-typing/coronavirus-research.html] kindly provided by Thermofisher Scientific, Victoria, Australia. In addition, two of the samples (GC-11 and GC-14) that yielded a low virus coverage by this method, were amplified separately (GC-11/38 and GC-14/37) using essentially the same method, but with the Ampliseq HiFi mix replaced with Amplitaq Gold 360 Master mix for the amplification step. Amplification was done for either 21, 27, or 35 cycles depending on the estimated virus load in the samples[26] and libraries prepared and run on Ion Torrent 530 chips on an Ion S5 XL genetic sequencer (Thermofisher Scientific) at a concentration of 50 pM as per the manufacturer's protocols[26,32,35]. Generated sequence reads were then mapped to a SARS-CoV-2 reference genome (Wuhan-Hu-1-NC_045512/MN908947.3 [https://www.ncbi.nlm.nih.gov/nuccore/NC_045512.2/ and https://www.ncbi.nlm.nih.gov/nuccore/MN908947.3]) using the TMAP software included in the Torrent Suite 5.10.1[36], and virus genomic consensus sequences generated using additional Torrent Suite plugins supplied by Thermofisher Scientific, and visualized in Integrative Genomic Viewer[37] (IGV 2.6.3) (Broad Institute, Cambridge, MA, USA). Near complete and partial SARS-CoV-2 genomes were aligned using Clustal-W[38] in MEGA 7 software[39] and near full length sequences (Accessions EPI_ISL_420855, EPI_ISL_420876-420877 and EPI_ISL_430064-430066)[26] submitted to the EpiCoV™ database of the Global Initiative on Sharing All influenza Database (GISAID)[40,41] (https://www.gisaid.org/) and to NCBI GenBank accession numbers MW192766, MW192771, MW192772, MW193406, MW193407, MW193408.

**Detection of SARS-CoV-2 subgenomic mRNAs in the NGS reads**. Although the SARS-CoV-2 Ampliseq panel used for the NGS has been designed to generate near full length SARS-CoV-2 genomic sequences, it uses simultaneous amplification of sample cDNA with a total of 242 primer pairs of which 237 primer pairs cover the near full genome of SARS-CoV-2 and an additional 5 amplicons targeting cellular genes in two primer pools[26]. Close inspection of all primers included in the panel, indicated that two of the forward primers (specifically the first forward primer in each of primer pool 1 and 2, see Thermofisher Scientific [https://www.thermofisher.com/au/en/home/life-science/sequencing/dna-sequencing/microbial-sequencing/microbial-identification-ion-torrent-next-generation-sequencing/viral-typing/coronavirus-research.html] and Source Data file for details) have their 3'-end at SARS-CoV-2 (NCBI Accession Wuhan-Hu-1-NC_045512/MN908947.3 [https://www.ncbi.nlm.nih.gov/nuccore/NC_045512.2/ and https://www.ncbi.nlm.nih.gov/nuccore/MN908947.3])[2] nucleotide 42 and 52, respectively, and consequently have a perfect match to a sequence included in the SARS-CoV-2 leader sequence with an estimated 27 or 17 nucleotides downstream of these primers also being part of the leader sequence[7,9,10,16].

Consequently, we concluded that the Ampliseq panel used here would potentially also amplify SARS-CoV-2 subgenomic RNAs by amplification from these two forward primers together with the closest downstream primer included in the same Ampliseq primer pool. Although this was not evident when assembling full virus genome sequences[26], a close inspection of reads around expected subgenomic RNA Transcription Regulatory Sites (TRS)[10] indicated that a significant number of NGS reads may have been amplified from subgenomic RNAs rather than from virus genomic RNA. To analyse this in more detail, we first assembled an exploratory composite reference for remapping using the Torrent Suite T-map reanalyze function. This initial assembled reference consisted of a composite reference with one sequence containing the first 21,500 nucleotides of the SARS-CoV-2 reference genome used for the initial assembly (NCBI Accession (Wuhan-Hu-1-NC_045512/MN908947.3 [https://www.ncbi.nlm.nih.gov/nuccore/NC_045512.2/ and https://www.ncbi.nlm.nih.gov/nuccore/MN908947.3])[2] to map reads most likely corresponding to the virus genome while we in addition, assembled ten tentative subgenomic RNA sequences containing 28 nucleotides from the 3'end of the leader sequence (of which the first 11 nucleotides would be from the forward primer from primer pool 2, if not enzymatically removed by the NGS process) and this leader then followed by the assumed TRS and gene specific

**Table 6 Table showing the primers designed and used to detect specific targets in the SARS-CoV-2 genome.**

| Set | Primers | Sequence (5'->3') | Length | Region |
|-----|---------|-------------------|--------|--------|
| 1 | SARS-CoV2-Ampliseq-P1-18-41-F | TCCCAGGTAACAAACCAACCAACT | 24 | Leader |
|   | SARS-CoV-2-RP1-3-TRB-27531-27512 | AAATGGTGAATTGCCCTCGT | 20 | Orf7a |
| 2 | SARS-CoV-2-FP2-TRB-27401-27425 | TTATTCTTTTCTTGGCACTGATAAC | 25 | Orf7a |
|   | SARS-CoV-2-RP1-3-TRB-27531-27512 | AAATGGTGAATTGCCCTCGT | 20 | Orf7a |
| 3 | SARS-CoV2-Ampliseq-P1-18-41-F | TCCCAGGTAACAAACCAACCAACT | 24 | Leader |
|   | SARS-CoV-2_RP4_206_187 | GACGAAACCGTAAGCAGCCT | 20 | 5' UTR |
| 4 | SARS-CoV-2_FP4_79-99 | AAAATCTGTGTGGCTGTCACT | 21 | 5' UTR |
|   | SARS-CoV-2_RP4_206_187 | GACGAAACCGTAAGCAGCCT | 20 | 5' UTR |
| 5 | SARS-CoV-2-FP2-TRB-27401-27425 | TTATTCTTTTCTTGGCACTGATAAC | 25 | Orf7a |
|   | SARS-CoV-2-RP1-2-TRB-27511-27491 | ATGTTCCAGAAGAGCAAGGTT | 21 | Orf7a |

sequence for the next 72 nucleotides. Consequently, this reference contained the first 21,500 nucleotides of the virus genome (Wuhan-Hu-1-NC_045512/ MN908947.3 [https://www.ncbi.nlm.nih.gov/nuccore/NC_045512.2/ and https:// www.ncbi.nlm.nih.gov/nuccore/MN908947.3]) as well as 10 composite references corresponding to the assumed 5'-end of the ten potential subgenomic RNAs; S, Orf3a, E, M, Orf6, Orf7a, Orf7b, Orf8, N, and Orf10/15[7,9,10]. This initial analysis indicated that this was an efficient way of mapping reads corresponding to subgenomic RNAs, and for our final analysis we updated the subgenomic RNA sequences in this composite reference to include the full leader sequence from nucleotide 1–69 and extended the gene specific sequences to ensure that they would include a reverse primer from each primer pool without extending into the next specific gene sequence. This final composite reference used for mapping then included the first 21,500 nucleotides of the SARS-CoV-2 genome and the ten subgenomic RNA specific sequences, each including the leader and gene specific sequences and having a length of 233–364 nucleotides (fasta file available as Supplementary Data 1). Mapped reads were visualized in IGV at a minimal alignment score of 60 and a mapping quality (MAPQ) of 84 and the abundance of reads mapped specifically to each subgenomic RNA at this stringency assessed by recording the read coverage at nucleotide position 61 of the leader sequence. Violin plots of the number of subgenomic reads were created using python version 3.8.5 and the libraries Matplotlib 3.1.2 [https://matplotlib.org/] and Seaborn 0.10.0 [https://seaborn.pydata.org/].

The next step in the analysis was to look at whether such subgenomic RNAs could be detected by a somewhat different type of analysis based on using reads already mapped to the full virus sequence, and then filter these to only look at reads containing part of the leader sequence. This type of analysis should give an unbiased view as to where reads containing the leader may be located on the genome, whether those sites correspond to the assumed position of the genomic leader and proposed TRS of the leader-containing subgenomic RNAs, whether the abundance somewhat correspond to that found by the method mentioned above and finally, whether any additional subgenomic RNAs or cryptic TRS sites may be detected. Reads within the mapped reads BAM files were filtered on whether they had a MAPQ of 32 or higher and contained the partial leader sequence GTAGATCTGTTCTCT, using a custom script (available at [https://github.com/ achamings/SARS-CoV-2-leader/tree/main]) written in BASH 4.4.20 and AWK 4.1.4 (GNU project, www.gnu.org) using samtools 1.7[42]. This sequence corresponded to nucleotides 52–67 within the SARS-CoV-2 leader sequence in GenBank sequence Wuhan-Hu-1-NC_045512/MN908947.3 [https://www.ncbi. nlm.nih.gov/nuccore/NC_045512.2/ and https://www.ncbi.nlm.nih.gov/nuccore/ MN908947.3][2]. Reads with this sequence immediately upstream from the mapped region of the read, or within the 5' end of the mapped region, were retained. The script then generated a spreadsheet giving the nucleotide position of where the leader sequence of each read finished in relation to the reference genome. These reads were then inspected in IGV. To assign the reads to the corresponding subgenomic RNA, the reads were grouped by the nucleotide position at the end of the leader sequence and tallied in Excel. Based on the end position of the leader mapped to the reference genome, each read was assigned to the corresponding subgenomic TRS. Typically, the leader sequence sat within a soft-clipped portion of each read, although depending on the reference sequence, the Ion Torrent TMAP algorithm did occasionally include the start of the leader sequence within the mapped portion of some reads, and at times included spurious insertions or deletions within this section of the mapping in its attempt to map the leader to the reference. Therefore, any read with the leader ending within 10nt of the start of the known subgenomic TRS sequences were assigned to the respective TRS. Some reads did not map to any known TRS, and these were assigned to an "Unknown TRS". Violin plots of the number of subgenomic reads were created as described above.

The next step in our analysis included searching the NCBI SRA from where we selected a few deposited NGS reads from studies employing the same SARS-CoV-2 Ampliseq panel used by us and in addition selected a few generated by different methods. SAM files from 15 SRA accessions were downloaded with the NCBI SRAtoolkit [https://trace.ncbi.nlm.nih.gov/Traces/sra/sra.cgi? cmd=show&f=software&m=software&s=software] sam-dump 2.8.2 and mapped

to the Ampliseq SARS-CoV-2 reference MN908947.3 [https://www.ncbi.nlm.nih. gov/nuccore/MN908947.3][2] using NCBI Magic-BLAST 1.3.0 with a minimum alignment score of 50 and percentage identity of 90% or higher. The script and analysis method to identify reads containing the leader sequence described above was used on the Magic-BLAST mapped SAM files for each of the SRA archives, and the number of reads corresponding to the start of each subgenomic RNA tallied.

**Abundance analysis of SARS-CoV-2 and cellular gene control amplicons.** To further look at generated NGS reads, we assessed the abundance of individual amplicons and the mapping data in more details. It should be mentioned, that processing of NGS reads by the Torrent Suite server initially includes trimming of barcode adapters and removal of low-quality and polyclonal reads. A base calling Phred score reflecting the signal quality at each base is then assigned and reads which have poor quality 3' ends are trimmed by scanning using a 30nt window until the average base calling quality drops to 15. Very short reads still remaining after this step (8 nucleotides or shorter), are then subsequently also removed. Consequently, all read numbers mentioned in this study are reads that have already satisfied these criteria and remained for further analysis/mapping.

We then checked reads mapped to the SARS-CoV-2 Ampliseq panel employed for the NGS that uses simultaneous amplification of sample cDNA with a total of 242 primer pairs of which 237 primer pairs cover the near full genome of SARS-CoV-2 and an additional 5 amplicons targeting cellular genes (see below and Thermofisher for additional details). Checking of mapped reads indicated that they were all mapped uniquely and thus counted only once. Counting/abundance of reads mapped to the individual amplicons were then done using BEDTools Version 2.27.1[43] with a minimum mapping quality (MAPQ) of 20 and requiring the mapped reads to cover more than 90% of an amplicon, and for the amplicons to cover no less than 70% of the read to be included in the count. This ensured reads were not counted more than once, as amplicons targeted partially overlapping regions of the SARS-CoV2 genome, and some of the smaller virus amplicons, were completely overlapped by a larger amplicon. This was done on all samples and in the same way for all the included 237 SARS-CoV-2 amplicons and the five control gene amplicons included in the Ampliseq panel. The five control gene amplicons span an intron of each of the following cellular genes, and thus amplify mRNAs for TATA-box binding protein (TBP NM_003194), LDL receptor related protein 1 (LRP1 NM_002332), hydroxymethylbilane synthase (HMBS NM_000190), MYC proto-oncogene (MYC NM_002467) and integrin subunit beta 7 (ITGB7 NM_000889). These control cellular gene amplicons are part of the Thermofisher Ampliseq^TM panel, and are automatically mapped as part of the SARS-CoV-2 mapping on the Ion Browser as described above.

**SARS-CoV-2 PCR assays to detect specific targets.** We designed primers for specific detection of the 7a subgenomic RNA by creating a forward primer in the leader sequence and a reverse primer within the 7a sequence itself. A second PCR targeting the Orf7a (i.e., both primers sitting within the 7a open reading frame and consequently detecting any RNA from full length SARS-CoV-2 genomic RNA as well as the subgenomic RNAs of S, Orf3, E, M, Orf6, and Orf7a) was also developed. Two PCRs specifically targeting the 5'-UTR were developed, one including part of the leader sequence and the other targeting the 5'-UTR downstream of the leader sequence. These two assays were specifically designed to only detect SARS-CoV-2 genomic RNA and not subgenomic RNAs. The primers are listed in Table 6.

These PCR assays were all performed using the same cDNA preparations as those used for the NGS; however, as we had limited cDNA volumes remaining for most samples, cDNA was diluted 2.5-fold except for sample GC13/35 for which we had more cDNA available and used undiluted cDNA. The PCRs all employed 2 μl of cDNA and 1× AmpliTaq Gold 360 PCR Mix, 1 μM of each primer, 2 μM Syto 9 (Thermofisher Scientific, Victoria, Australia) and a PCR protocol of 95 °C for 10 min, 40 cycles of 95 °C for 30 s, 58 °C for 30 s, 72 °C for 30 s, and a final 72 °C step for 3 min. A melt curve analysis was performed immediately post PCR with the reaction conditions of 95 °C for 15 s, then 60 °C for 1 min followed by a continuous temperature ramp between 60 °C and 95 °C increasing at 0.05 °C/s. Positive results

were called based on threshold cycle and the correct peak melt temperature of the product. For the initial assay set up, amplicon identity was further confirmed by gel electrophoresis followed by Sanger sequencing of the PCR products[32,44].

In addition to the in-house PCR assays described above, we also used the commercial TaqPath™ COVID-19 RT-PCR Kit (Thermofisher Scientific, Victoria, Australia) using 1.5 µl of the same diluted cDNA samples mentioned above (except for sample GC-13/35 for which we had more cDNA available and used 2.5 µl of undiluted cDNA) and employing the TaqPath™ 1-Step Multiplex Master Mix without ROX (Thermofisher Scientific, Victoria, Australia) together with the TaqPath™ COVID-19 RT-PCR Kit (Thermofisher Scientific, Victoria, Australia)[26] although skipping the initial reverse transcription step. This assay simultaneously detects three targets; a target in the Orf1 only detecting the virus genomic RNA, a target in the S gene detecting genomic and S subgenomic RNA and a target in the N gene detecting genomic RNA as well as all full length subgenomic RNAs except Orf10 if present. Samples were identified as having a high or low virus load based on the Ct obtained from the COVID-19 RT-PCR kit assay or the Ct reported from the original diagnostic laboratory[26].

Efficiency, slope and theoretical sensitivity of each PCR were determined by using a dilution series of gel purified amplicons for the in-house assays and a dilution series of a positive control included in the commercial COVID-19 kit. For in house assays using either Syto 9 or SYBR Green (see below), the amplification slopes of the assays were very similar with around 3.9-4.0 cycles between each 10-fold dilution and a lower Ct sensitivity/threshold of 30-32 while the commercial probe-based assay, as anticipated, was more sensitive and efficient with around 3.4 cycles between each 10-fold dilution and a lower Ct sensitivity/threshold of 39 for all three targets included.

**Strand specific PCR.** For strand specific PCR detection, we used the original nucleic acids extracted for NGS and using an initial step to denature any double-stranded RNA by first heating at 95 °C for 3 min followed by snap-freezing at −20 °C. The samples were then tested using real-time SYBR Green PCR assays with the Power SYBR Green RNA-to-CT 1 step kit (Applied Biosystems, California, USA) using the 7a PCRs described above and adapted so that we initially added a single primer for the reverse transcription step of the protocol at 48 °C for 30 min, then inactivated the reverse transcription enzyme by incubation at 95 °C for 8 min before adding the other primer and continuing the protocol by initially heating to 95 °C for 2 min to further activate the PCR enzyme before conducting 40 cycles of 95 °C for 30 s, 58 °C for 30 s, 72 °C for 30 s and a final 72 °C step for 3 min. A melt curve analysis was performed and positive results as described above. The initial assay set up was further confirmed by gel electrophoresis of the products followed by Sanger sequencing to confirm amplicon identity[32,44].

To analyse if samples testing positive for minus sense SARS-CoV-2 RNA potentially contained double stranded RNA, we attempted treatment of samples with RNase If (New England Biolabs (NEB), Victoria, Australia) to preferentially remove single stranded RNA before PCR. This was performed essentially as described[45], but with a slight modification to promote annealing of extracted RNA before digestion. This was done by adding 1/10th volume of the 10 × RNase If buffer (NEBuffer 3) and incubating at room temperature for 10 min before adding 50 units of RNase If and then incubating at 37 °C for 10 min, heating at 95 °C for 3 min followed by snap-freezing at −20 °C to heat inactivate the enzyme and denature any double-stranded RNA before PCR.

**Membrane association and nuclease resistance of SARS-CoV-2 RNAs.** To study a potential membrane association and nuclease resistance of SARS-CoV-2 RNAs, we modified a protocol described for analysis of SARS-CoV replication/transcription complexes in cell culture[23]. This protocol was followed with the following minor modifications. To allow analysis of swab sample material that had already been frozen and thawed at least twice, we started the protocol without an initial Dounce homogenizer step and starting with the swab material in PBS without any additional chemicals or RNase inhibitors. The first step of fractionation consisted of centrifugation at 1000 × g for 5 min and taking the pellet (designated P1, including approximately 10% of the volume of the original sample) and the supernatant fraction (designated S1, approximately 90% of the original volume). The P1 and S1 fractions were then each divided into two aliquots, of which one was treated with 0.5% of the nonionic detergent Triton X-100 for 15 min at 4 °C. These fractions were then again split into two aliquots of which one was treated with nucleases, first adding a 20 × nuclease buffer and then benzonase and micrococcal nuclease and incubation at room temperature for 30 min[24,32]. Fractions were then centrifuged at 10,000 × g for 10 min and the pellet fraction (designated the P10 fraction) and the supernatant (designated S10) collected. In effect, this resulted in a total of 16 fractions from each sample of which 8 came from each of the P1 and S1 fractions and of which half had been treated with Triton and the other half not, and then half of these fractions treated with nucleases or not. Final fractions were designated P1P10, P1S10, S1P10 and S1S10 and including aliquots that had been treated or not with Triton (T+ or T−) and treated or not with nuclease (N+ or N−). These fractions were then subjected to nucleic acids extraction and cDNA preparation as described for NGS, and tested by the 7a subgenomic and genomic PCRs as well as the commercial kit as described above. The obtained PCR values were normalized to the final volume of sample in each of the fractions in order to compare the results. In addition, we also tested the nucleic

acids from these fractions in the strand-specific PCRs for 7a genomic and subgenomic RNA.

**Reporting summary.** Further information on research design is available in the Nature Research Reporting Summary linked to this article.

## Data availability
The sequence reads of our SARS-CoV-2 positive samples reported here have been deposited in the NCBI Sequence Read Archive (SRA) under SRA accession: PRJNA636225. The fasta file used for mapping are available as Supplementary Data 1. Assembled near full length SARS-CoV-2 genomic sequences are deposited in GISAID [https://www.gisaid.org/]; Accessions EPI_ISL_420855, EPI_ISL_420876-420877 and EPI_ISL_430064-430066) and in NCBI GenBank accession numbers MW192766, MW192771, MW192772, MW193406, MW193407, and MW193408. Other assembled nucleotide sequences or nucleotide sequence read archives mentioned are publicly available at NCBI [https://www.ncbi.nlm.nih.gov/nuccore/ and https://www.ncbi.nlm.nih.gov/sra/]. All other data supporting the findings of this manuscript are available in the Supplementary Information files or from the corresponding author upon request. Source data are provided with this paper. A reporting summary for this Article is available as a Supplementary Information file.

## Code availability
Code used to identify reads containing the leader sequence is available at: [https://github.com/achamings/SARS-CoV-2-leader/tree/main].

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

## Acknowledgements

We gratefully acknowledge clinical and laboratory diagnostic staff for providing samples and for doing the initial diagnostic SARS-CoV-2 testing. We acknowledge Thermofisher Scientific, Victoria, Australia, for supplying the Ampliseq panel used. We also acknowledge Jason Hodge, laboratory manager of the GCEID laboratory for his technical input. Finally, we gratefully acknowledge the authors and originating and submitting laboratories for the sequence read archives we have used in this study (Supplementary Table 3). This research was funded by Deakin University, Barwon Health and CSIRO and from the National Health and Medical Research Council (NHMRC) equipment grant number GNT9000413 to S.A.

## Author contributions

S.A. initiated the study, coordinated all work carried out at GCEID and did the initial mapping of SARS-CoV-2 subgenomic RNAs. A.C. did the script-based mapping, mapping of selected SRAs and did the fractionation experiment. T.R.B. provided sample information, participated in the NGS and initial analysis to look for subgenomic RNAs, and did the targeted and strand-specific PCRs. S.A. drafted the initial manuscript with inputs from A.C. and T.R.B. All authors contributed to the final submitted version. All authors have read and agreed to the final version of the manuscript.

## Competing interests

The authors declare no competing or conflict of interest. The funders had no role in the design of the study; in the collection, analyses, or interpretation of data; in the writing of the manuscript, or in the decision to publish the results.
