## [Peer Review File · Nature Communications]

REVIEWER COMMENTS, first round -

Reviewer #1 (Remarks to the Author):

COVID-19 is a novel disease and our understanding on this disease is rather limited at this stage. Several clinical studies have indicated that COVID-19 patients at very late disease onset can have recurrent positivity of SARS-CoV-2 RNA. The reason account of this observation is yet to be fully explained.

In this study, the authors attempt to explain this by using the findings deduced from their next generation sequencing analyses. In short, the authors conclude that this might be due to the presence of subgenomic mRNA molecules that are protected double membrane vesicles in these "relapsed samples". Although the hypothesis is interesting, the reviewer has the following concerns:

Major:

1. The study entirely a sequence analysis study. No work has been done to prove that these subgenomic mRNA is protected by double membrane vesicles in the clinical samples. Some experimental evidences for this are needed.
2. No RT-PCR has been done to demonstrate the prolonged or recurrent RT-PCR positivity is due to these RNA species.
3. Their interpretations might not necessary reflect the data presented in this study. For example, the reviewer finds the subgenomic mRNA of ORF8 is much less than those of ORF7 and N (Figs and tables, but this is not what they have reported in the main text (page 5)).

Minor:

1. The tables and figures are highly redundant.
2. No basic information about the studied clinical samples (e.g. day of disease onset). The reviewer finds it is very hard to relate their findings to clinical observations from other studies.
3. This reviewer appreciates that there are paired samples collected about 2 weeks apart. But the authors should aware that there are patients who can shed viral RNA for many weeks. Do the authors suggest that these membrane-bound RNA species can be kept in the patients for such a long time?
4. This paper is presenting a simple concept. The manuscript can be presented in a much more concise manner. It is unlikely unnecessary to have that many speculations.

Reviewer #2 (Remarks to the Author):

Following the criteria in instructions to reviewers:

- What are the noteworthy results?

This paper uses NGS to sequence SARS-CoV-2 RNAs from clinical samples obtained from infected individuals. They focus on the prevalence of the many subgenomic mRNAs generated by the virus in the cell during virus infection, but which are not thought to be encapsidated in significant numbers. The authors can distinguish between 5' ends of sgRNAs, and reads from the homologous region in genomic RNA due to the presence of the genomic 5' UTR on the end of each sgRNA, joined at the conserved transcriptional regulatory sequence (TRS). Thus, reads that span this kind of "splice junction" reveal the specific sgRNA. The authors could have explained this more clearly to the more general audience of this journal who may not know about the structure of coronavirus sgRNAs, by including a figure showing a map of the genome and sgRNAs with 5' leader and TRS sequences, primer binding sites indicated.

The authors find large variations in number of reads of sgRNAs detected between patients, and within a sample for the different sgRNAs. They found that the poor quality samples gave a higher ratio of apparent sgRNA to genomic RNA, but speculated that this was because the degraded RNA

would yield fewer PCR products because the primers are farther apart. This makes sense and is not surprising. Also, the authors show they get similar results using different polymerases for PCR amplification. The authors also compared their results with reads in the NCBI Sequence Read Archive (SRA).

- Will the work be of significance to the field and related fields?

The work is of some interest to virologists investigating sgRNA levels in infected patients. Because it involves SARS-CoV-2 it is significant of course, but it isn't clear from the data presented how this will provide a major improvement in our understanding of the virus or clinical treatments.

How does it compare to the established literature?

As SARS-CoV-2 is a new virus, to my knowledge these results showing variation among samples from patients are new. However far more extensive and detailed analysis of SARS-CoV-2 sgRNAs have been done. For example see reference 8: Chang et al. Cell 2020 doi: 10.1016/j.cell.2020.04.011 That paper focused on cultured cells, so the work submitted here is of interest in that it uses RNAs from clinical samples. The authors could have used that paper as model for how they might have described experiments and analysed and displayed data in more highly informative ways. Also, it is not surprising that, for example, the N protein-coding sgRNA is highly abundant, based on studies of other coronaviruses. However, it is interesting that no sgRNA was detected for ORF 10 or ORF 7b.

- Does the work support the conclusions and claims, or is additional evidence needed?

Not all claims are supported. The proposed role of double membrane vesicles is pure speculation (see below). The amount of sgRNAs vary so much between individuals samples, it's hard to draw solid conclusions about the abundance of all but the extremely high and extremely low abundance sgRNAs. For that matter statistical analysis to determine the significance of the differences in sgRNA levels (reads) is lacking.

- Are there any flaws in the data analysis, interpretation and conclusions? - Do these prohibit publication or require revision?

See comments above.

The authors interpret their results as indicating sgRNAs are associated with double membrane vesicles in which the virus has been shown to replicate by others. The authors provide no evidence for this. There is no visualization of RNAs in cells or isolation of membrane vesicles or other types of cell biology to support this claim.

Also, two individuals were sampled twice (11 and 17 days apart). The authors found greater levels of sgRNAs in the second samples for both. They conclude that this means sgRNAs increase at later stages of infection, but it is possible that at both timepoints, groups of individual cells at all different stages of infection were sampled. These are nasal and oral swabs, not synchronous infections. So we don't know what stage(s) of infection were sampled.

- Is the methodology sound? Does the work meet the expected standards in your field?

The NGS and analysis of sequence data appears to be sound.

- Is there enough detail provided in the methods for the work to be reproduced?

No. Many times throughout the Methods, authors refer to methods that are in another manuscript that has only been submitted for publication.

Reviewer #3 (Remarks to the Author):

General review

What are the major claims of the paper?

The paper describes detection of SARS-CoV-2 subgenomic RNAs in routine diagnostic oropharyngeal/nasopharyngeal swabs subjected to next generation sequencing (NGS – Ion

Torrent). They found that subgenomic RNAs are present in most samples, but that the overall, and individual, abundance varies among samples and may be related to stage of infection and, importantly, more related to how samples were taken and treated before testing/sequencing. The authors claim that their specific detection of subgenomic RNAs in clinical samples indicates that these RNAs are rather stable and most likely found in, and protected by, membrane structures.

In addition they claim that detection of subgenomic RNAs in clinical samples, importantly, do not necessarily signify active virus replication/transcription, but instead is due to such RNAs being part of double-membrane vesicles and thus relatively stable compared to cellular mRNA.

Are they novel and will they be of interest to others in the community and the wider field?

There is a novelty in their claims and it could be of interest to the community.

Is the work convincing, and if not, what further evidence would be required to strengthen the conclusions?

The work is not convincing. For instance they claim that the subgenomic regions are more stable than cellular RNA, yet this is not directly proved in this work i.e. I would like to see that they are more stable than other parts of the virus (orf1a and orf1b) as well as positive controls of human genes expressed in the samples. To my understanding both of the above exist in the amplicon kit used. Also, as an example for a control see the use of ABL1 gene in the work of Ishige et al., Clin Chim Acta. 2020 Aug; 507: 139–142.

They explain that the subgenomic regions stability has two causes, RNA degradation in vitro or as our part of a biological process in vivo. It is possible with proper experimental design to distinguish between these two (as explained above using controls).

I find it problematic that the stability issue is not established in this work in a statistically quantitative manner.

It will be beneficial to add a coverage plot depicted the coverage on the full genome for all the samples with sufficient coverage.

Questions and concerns about the paper.

Why do the authors think there is a difference in the stability of the various subgenomic regions? The connection between the disease state (days after initial detection of infection) and the expression level of the subgenomics regions is not well established, there is a need for more samples, statistics and clinical information of the disease state.

We would also be grateful if Authors could comment on the appropriateness and validity of any statistical analysis, as well the ability of a researcher to reproduce the work, given the level of detail provided.

Questions and remarks:

Methods section lacks information:

1. Were there any QC measures applied to the sequences i.e. filtering reads by quality or length?
2. How was the quantification of virus genes done? How did Authors deal with the reads multiple aligned? This is a critical step due to the leader sequence that is shared among the subgenomic regions.

3. Please show how many of the reads align to SARS-CoV-2 reference genome as a whole (uniquely and not uniquely)

4. Please show the number of reads that specifically align to the 21,500 bases of the genome as well at the human controls present.

5. See the paragraph below copied from methods.

This final composite reference used for mapping then included the first 21500 nucleotides of the SARS-CoV-2 genome and the 10 subgenomic RNA specific sequences, each including the leader and gene specific sequences and having a length of 233-364 nucleotides (Supplementary Information S1 [file: Wuhan-Hu-1-NC_045512-21500-and-subgenomics-SA4.fasta]). Mapped reads were visualised in IGV at a minimal alignment score of 60 and a mapping quality (MAPQ) of 84.

No coverage IGV plot is presented.

6. It is recommended to provide the bam files available as well as the genome used to map.

7. Authors write "56 million NGS reads generated from the 14 virus-positive samples, nearly 800,000 reads mapped to one of the 10 SARS-CoV-2 subgenomic RNAs" In other words the whole analysis here is to 1.4% of the reads. Need to explain why is this number so low.

8. The number of reads that align the whole genome should be used for normalization between the samples. The way Authors performed normalization is not clear.

Results

1. Table 1 (and all other tables and plots) should contain counts to orf1a, orf1b and the human controls.

2. Please explain what is this full virus genome? Looking at Authors previous article there are several genomes.

3. Supplementary Figure S5: Average coverage per 5 million reads for samples 37, 38, 60, 61, 62 and 63

This plot has repeated measures of two people yet it is not easy to follow which is a repeated measure of whom.

4. In general the only plots presented are histograms. Yet, they are not a good choice since the authors try to demonstrate two issues here:

- a. There is a difference within a certain sample between the expressions of the various regions.
- b. There is difference between the samples "profile", since some samples are more degraded or from a later stage in the disease.

This can be demonstrated using clustering among the samples and a heat map of standardized values.

5. Title - Detection of subgenomic RNAs mapped to the virus genome by filtering reads containing the partial leader sequence

The word 'the' should be emitted.

6. Authors write "only barcodes" , this is the first mention of barcodes, perhaps authors mean samples.

Discussion

1. Authors write "Two different approaches used", need to explain what they are.

2. Authors write "as we believe the study described by Zhang et al. is not mapping subgenomic RNAs but simply reports coverage for the different parts of the virus genome." What do Authors mean?

3. Authors write; "we present information that helps understand prolonged and sometimes

inconsistent PCR-positivity and may pave the way for development of better diagnostic PCRs”
How exactly are the authors proposing to improve the diagnosis and the PCR?

The reviewers have raised some important points regarding the need for a more detailed description of the quality control and analysis of our NGS data, experimental validation of our hypothesis regarding stability of SARS-CoV-2 sgRNA and membrane association/protection and to provide more information regarding the patient samples. We thank the reviewers for their helpful suggestions and believe we have addressed the reviewers' concerns in full in our significantly revised version of our manuscript.

Below we give a detailed point-by-point response (*in blue text for clarity only*) to each of the items mentioned by the reviewers and all changes in the manuscript text file are shown with track changes. We have also released our NGS data in the NCBI SRA to ensure that reviewers are able to access all data.

REVIEWER COMMENTS

Reviewer #1 (Remarks to the Author):

COVID-19 is a novel disease and our understanding on this disease is rather limited at this stage. Several clinical studies have indicated that COVID-19 patients at very late disease onset can have recurrent positivity of SARS-CoV-2 RNA. The reason account of this observation is yet to be fully explained.

In this study, the authors attempt to explain this by using the findings deduced from their next generation sequencing analyses. In short, the authors conclude that this might be due to the presence of subgenomic mRNA molecules that are protected double membrane vesicles in these "relapsed samples". Although the hypothesis is interesting, the reviewer has the following concerns:

Major:

1. The study entirely a sequence analysis study. No work has been done to prove that these subgenomic mRNA is protected by double membrane vesicles in the clinical samples. Some experimental evidences for this are needed.

We have added a section to the manuscript on "Membrane association and nuclease resistance of SARS-CoV-2 RNAs" with results obtained using a method described for SARS-CoV to fractionate coronavirus transcription complexes in membrane vesicles. We are able to show that the SARS-CoV-2 RNAs detected in diagnostic samples are found in fractions

expected to contain transcription complexes and to a large extent are protected from nucleases and that this protection is no longer present when samples are treated with detergent.

2. No RT-PCR has been done to demonstrate the prolonged or recurrent RT-PCR positivity is due to these RNA species.

We have included results “SARS-CoV-2 PCR assays to detect subgenomic 7a RNA, genomic and subgenomic 7a RNA and genomic only 5'-UTR RNA” using a PCR to specifically detect and semi-quantify the leader-containing 7a subgenomic RNA and use this for comparison to results from other PCRs developed to detect either the 7a genomic and all subgenomic RNAs up to and including the 7a subgenomic RNA or to detect the 5-UTR of the genomic RNA only. In addition, we take one step further “Strand specific PCR” and use these PCRs to also attempt to detect the negative strand of these RNAs.

3. Their interpretations might not necessary reflect the data presented in this study. For example, the reviewer finds the subgenomic mRNA of ORF8 is much less than those of ORF7 and N (Figs and tables, but this is not what they have reported in the main text (page 5). We thank the reviewer for spotting this, it is caused by an error in the sentence and we gave revised the text (page 5) so that the order reflects the data shown in the Table and the new Figure 1. In addition, the manuscript has been updated with the data obtained by PCR mentioned above and now includes a much more extensive comparison of read abundances as compared to PCR results.

Minor:

1. The tables and figures are highly redundant.

Both Tables and Figures have been extensively revised and the number of Figures reduced.

2. No basic information about the studied clinical samples (e.g. day of disease onset). The reviewer finds it is very hard to relate their findings to clinical observations from other studies.

We have now included a new Table 1 that gives summary details about the individuals and samples included in our study.

3. This reviewer appreciates that there are paired samples collected about 2 weeks apart. But the authors should aware that there are patients who can shed viral RNA for many weeks. Do the authors suggest that these membrane-bound RNA species can be kept in the patients for such a long time?

We are aware of positive detection of SARS-CoV-2 RNA for several weeks; we have extended the results and discussion around our findings as well as those of others, and believe that we are able to show that membrane-bound RNAs are likely to be found in infected individuals for several weeks, however, as the amount of these RNAs overall are in a lower amount than virion RNA, detection limits are reached earlier. We refer to the revised manuscript for further explanation and details.

4. This paper is presenting a simple concept. The manuscript can be presented in a much more concise manner. It is unlikely unnecessary to have that many speculations.

As we have added additional results using both PCR and fractionation together with detergent and nuclease treatment, and in addition have revised the manuscript significantly, we believe that it contains less speculation and describe the various findings in context.

Reviewer #2 (Remarks to the Author):

Following the criteria in instructions to reviewers:

- What are the noteworthy results?

This paper uses NGS to sequence SARS-CoV-2 RNAs from clinical samples obtained from infected individuals. They focus on the prevalence of the many subgenomic mRNAs generated by the virus in the cell during virus infection, but which are not thought to be encapsidated in significant numbers. The authors can distinguish between 5' ends of sgRNAs, and reads from the homologous region in genomic RNA due to the presence of the genomic 5' UTR on the end of each sgRNA, joined at the conserved transcriptional regulatory sequence (TRS). Thus, reads that span this kind of "splice junction" reveal the specific sgRNA. The authors could have explained this more clearly to the more general audience of this journal who may not know about the structure of coronavirus sgRNAs, by including a figure showing a map of the genome and sgRNAs with 5' leader and TRS sequences, primer binding sites indicated.

We do not feel that it is appropriate for us to add a Figure of the structure of coronavirus subgenomic RNAs. These subgenomic RNAs are well documented in the references we provide e.g. in the Introduction, and clearly, we do not define these subgenomic RNAs as such, but rather map our NGS reads from diagnostic samples based on what is already well known and published from cell culture and then subsequently, validate our findings by using a second mapping method that is essentially unbiased to confirm our initial findings. Consequently, we believe that readers are able to easily find such a Figure in the references given.

The authors find large variations in number of reads of sgRNAs detected between patients, and within a sample for the different sgRNAs. They found that the poor quality samples gave a higher ratio of apparent sgRNA to genomic RNA, but speculated that this was because the degraded RNA would yield fewer PCR products because the primers are farther apart. This makes sense and is not surprising. Also, the authors show they get similar results using different polymerases for PCR amplification. The authors also compared their results with reads in the NCBI Sequence Read Archive (SRA).

- Will the work be of significance to the field and related fields?

The work is of some interest to virologists investigating sgRNA levels in infected patients. Because it involves SARS-CoV-2 it is significant of course, but it isn't clear from the data presented how this will provide a major improvement in our understanding of the virus or clinical treatments.

We believe that the understanding of the presence and stability of subgenomic RNAs in diagnostic samples is important. This is not because it directly provides improvement in clinical treatment, however, it is important because an early paper, referenced in our study, assumed that the presence of subgenomic RNA was correlated with early/active infection and this assumption has been taken up by a number of later studies, now also referenced in the revised manuscript. We believe we have shown in the revised manuscript that these subgenomic RNAs are highly stable, possibly as stable as virion RNA, but due to the fact that they are found in a lower quantity, falls to below detection limits in PCR earlier than virion

RNA. Although we cannot formally show this in our data, this is evident in a preprint from a Dutch group to which we refer in the updated manuscript. That study quite clearly shows that subgenomic RNA can be found for a rather long period and in a rather consistent, lower, ratio as compared to virion RNA. In other words, our detailed studies on a relatively low number of samples are consistent with a less detailed study only using PCR, but which includes many more samples than we have available.

How does it compare to the established literature?

As SARS-CoV-2 is a new virus, to my knowledge these results showing variation among samples from patients are new. However far more extensive and detailed analysis of SARS-CoV-2 sgRNAs have been done. For example see reference 8: Chang et al. Cell 2020 doi: 10.1016/j.cell.2020.04.011 That paper focused on cultured cells, so the work submitted here is of interest in that it uses RNAs from clinical samples. The authors could have used that paper as model for how they might have described experiments and analysed and displayed data in more highly informative ways. Also, it is not surprising that, for example, the N protein-coding sgRNA is highly abundant, based on studies of other coronaviruses. However, it is interesting that no sgRNA was detected for ORF 10 or ORF 7b.

We believe that we do use and refer to the work done by Kim et al. (note Kim is the first author and Chang last author on that paper), however, their study is different as they look at cell culture and specifically look to define the SARS-CoV-2 subgenomic RNAs. We believe we have updated the way we show and describe our data, and clearly, even in the initial manuscript, referred to how our results from diagnostic samples compared to the findings in cell culture. Nevertheless, their study is very different as it looks at early infection in cell culture while we look at much later infection in routine diagnostic samples.

- Does the work support the conclusions and claims, or is additional evidence needed?

Not all claims are supported. The proposed role of double membrane vesicles is pure speculation (see below). The amount of sgRNAs vary so much between individuals samples, it's hard to draw solid conclusions about the abundance of all but the extremely high and extremely low abundance sgRNAs. For that matter statistical analysis to determine the significance of the differences in sgRNA levels (reads) is lacking.

We have extended the presentation and comparison of read/amplicon abundances in the revised manuscript. Furthermore, we compare the findings by NGS to the new data obtained by our newly developed PCRs. While direct statistical analysis is not possible due to the large variation of reads, we do compare various ways of comparing NGS read abundance and how that fits with the results obtained by PCR. Furthermore, we add results in regards to the strand-specificity of the RNAs detected and also look at abundance as well as detergent and nuclease sensitivity in fractions obtained by centrifugation based on a method described for SARS-CoV transcription complexes. Overall, we believe that we are able to show that albeit highly variable, the amounts of subgenomic RNAs are roughly in the amounts that would be expected, albeit with the caveat that diagnostic samples are more comparable to semi-purified virion preparations than to early intracellular RNA in infected cell cultures. This may not sound important, however, we think that the distinction is crucial, because as mentioned elsewhere, several studies have assumed that subgenomic RNAs are only present very early on in infection and then quickly disappear, which is in contrast to our findings and those in the preprint from the Dutch group mentioned above and in the paper.

- Are there any flaws in the data analysis, interpretation and conclusions? - Do these prohibit publication or require revision?

See comments above.

The authors interpret their results as indicating sgRNAs are associated with double membrane vesicles in which the virus has been shown to replicate by others. The authors provide no evidence for this. There is no visualization of RNAs in cells or isolation of membrane vesicles or other types of cell biology to support this claim.

As indicated above, we have included new results based on a method described for SARS-CoV transcription complexes and are able to show that the SARS-CoV2 RNAs we detect are highly protected from nucleases, fractionate into the fractions expected and that the nuclease protection is no longer present when samples are subjected to mild detergent treatment.

Also, two individuals were sampled twice (11 and 17 days apart). The authors found greater levels of sgRNAs in the second samples for both. They conclude that this means sgRNAs increase at later stages of infection, but it is possible that at both timepoints, groups of individual cells at all different stages of infection were sampled. These are nasal and oral swabs, not synchronous infections. So we don't know what stage(s) of infection were sampled.

In the revised version of the manuscript we present data on mapping of NGS reads to included cellular mRNA control amplicons and compare the levels of NGS reads to findings using PCR. We are also able to show and discuss, that samples of poor quality or with a low virus load may result in amplification of certain amplicons more than others. This is compared and discussed in much detail in the revised manuscript where we focus on comparing the overall ratios of the different RNAs/amplicons more than on any individual one as abundance of these amplicons, as mentioned by the reviewer, may differ from sample to sample.

- Is the methodology sound? Does the work meet the expected standards in your field?

The NGS and analysis of sequence data appears to be sound.

Thanks, we appreciate this comment.

- Is there enough detail provided in the methods for the work to be reproduced?

No. Many times throughout the Methods, authors refer to methods that are in another manuscript that has only been submitted for publication.

We have added substantial more detail about the NGS and the mapping and have also added a new Table 1 giving summary information about samples etc. and a Table 3 as well as a detailed Supplementary Table S2 with mapping information. Also, we have released the data in the NCBI SRA. Overall, we think that the revised manuscript is much improved and contains the needed details.

Reviewer #3 (Remarks to the Author):

General review

What are the major claims of the paper?

The paper describes detection of SARS-CoV-2 subgenomic RNAs in routine diagnostic oropharyngeal/nasopharyngeal swabs subjected to next generation sequencing (NGS – Ion Torrent). They found that subgenomic RNAs are present in most samples, but that the overall, and individual, abundance varies among samples and may be related to stage of infection and, importantly, more related to how samples were taken and treated before testing/sequencing.

The authors claim that their specific detection of subgenomic RNAs in clinical samples indicates that these RNAs are rather stable and most likely found in, and protected by, membrane structures.

In addition they claim that detection of subgenomic RNAs in clinical samples, importantly, do not necessarily signify active virus replication/transcription, but instead is due to such RNAs being part of double-membrane vesicles and thus relatively stable compared to cellular mRNA.

Are they novel and will they be of interest to others in the community and the wider field?

There is a novelty in their claims and it could be of interest to the community.

Is the work convincing, and if not, what further evidence would be required to strengthen the conclusions?

The work is not convincing. For instance they claim that the subgenomic regions are more stable than cellular RNA, yet this is not directly proved in this work i.e. I would like to see that they are more stable than other parts of the virus (orf1a and orf1b) as well as positive controls of human genes expressed in the samples. To my understanding both of the above exist in the of the amplicon kit used. Also, as an example for a control see the use of ABL1 gene in the work of Ishige et al., Clin Chim Acta. 2020 Aug; 507: 139–142.

They explain that the subgenomic regions stability has two causes, RNA degradation in vitro or as our part of a biological process in vivo. It is possible with proper experimental design to distinguish between these two (as explained above using controls).

I find it problematic that the stability issue is not established in this work in a statistically quantitative manner.

We do not believe that we claim that the detected subgenomic RNAs are more stable than e.g. virion RNA or cellular mRNA. However, we believe that the fact that we are able to detect such subgenomic RNAs for up to 17 days after initial detection, the latest time point available to us, is consistent with these RNAs being relatively stable. In any event, we are able to show, using a method originally published to study SARS-CoV transcription complexes in cell culture, that these RNAs are highly protected from nuclease degradation. Furthermore, in our revised manuscript we have now also included analysis of the cellular mRNA amplicons included in the NGS panel, and present the details of the abundance of cellular mRNA amplicons detected in the samples. Finally, we have also included PCR results in the revised manuscript.

It will be beneficial to add a coverage plot depicted the coverage on the full genome for all the samples with sufficient coverage.

We have not added a coverage plot; however, we have added another supplementary Table S2 that gives all details about number of reads mapped to individual amplicons.

Furthermore, we have released the data deposited in the NCBI SRA and thus all details about the reads and the reference used for mapping are available.

Questions and concerns about the paper.

Why do the authors think there is a difference in the stability of the various subgenomic regions?

We do not believe that we have stated that there is a difference in stability of the various subgenomic regions, we present the results we have and in the revised manuscript use this for comparison to results using PCR. The variability observed and how this may be compared in different ways are now included in the revised manuscript.

The connection between the disease state (days after initial detection of infection) and the expression level of the subgenomics regions is not well established, there is a need for more samples, statistics and clinical information of the disease state.

We are not sure what is meant here, however, the individuals included in our study had only mild disease and were not hospitalised but rather stayed in self-isolation. However, in the revised manuscript, we do refer to a preprint by a Dutch group, in which they look, using PCR for the E subgenomic RNA only, at a much larger group of hospitalised individuals with more severe disease, and they report detection of the E subgenomic RNA in such settings in up to 22 days after first clinical symptoms.

We would also be grateful if Authors could comment on the appropriateness and validity of any statistical analysis, as well the ability of a researcher to reproduce the work, given the level of detail provided.

Based on the levels of details now given including additional Tables as well as the NCBI SRA and reference used, we believe that researchers are able to reproduce the work; in that connection we would like to mention that in the manuscript we are also able to map such subgenomic RNAs in selected read archives. In regards to statistical analysis, we have given details of all mapping details and presented them in box and whiskers plots etc. We do not go into a direct statistical analysis of the data; however, in the revised manuscript we present several ways of comparing amplicon abundances and furthermore, compare findings using NGS with findings using PCR and fractionation including detergent and nuclease treatment.

Questions and remarks:

Methods section lacks information:

1. Were there any QC measures applied to the sequences i.e. filtering reads by quality or length?

We have added a section describing this under the new section “Further abundance analysis of SARS-CoV-2 amplicons and cellular gene control amplicons included in the Ampliseq panel”.

2. How was the quantification of virus genes done? How did Authors deal with the reads

multiple aligned? This is a critical step due to the leader sequence that is shared among the subgenomic regions.

We have added more information under “Further abundance analysis of SARS-CoV-2 amplicons and cellular gene control amplicons included in the Ampliseq panel” regarding the fact that reads were all uniquely mapped and furthermore, we have added a new Supplementary Table S2 with all details regarding mapping to specific amplicons and have also added information regarding mapping to included cellular control amplicons.

3. Please show how many of the reads align to SARS-CoV-2 reference genome as a whole (uniquely and not uniquely)

As mentioned above, we have included text to state that all reads were mapped uniquely and we have added Supplementary Table S2 with the number of reads mapped to the individual amplicons in the SARS-CoV-2 reference genome used.

4. Please show the number of reads that specifically align to the 21,500 bases of the genome as well as the human controls present.

As mentioned above, we have included a new Supplementary Table S2 with the number of reads mapped to the individual amplicons in the SARS-CoV-2 reference genome and this Table also give the number of reads mapped to amplicons in the first 21500 nucleotides of the genome. This Table also shows the number of reads mapped to the human cellular mRNA control amplicons and those numbers and in addition shown in a new Table 3 focusing on the number of reads mapped to cellular mRNA control amplicons.

5. See the paragraph below copied from methods.

This final composite reference used for mapping then included the first 21500 nucleotides of the SARS-CoV-2 genome and the 10 subgenomic RNA specific sequences, each including the leader and gene specific sequences and having a length of 233-364 nucleotides (Supplementary Information S1 [file: Wuhan-Hu-1-NC_045512-21500-and-subgenomics-SA4.fasta]). Mapped reads were visualised in IGV at a minimal alignment score of 60 and a mapping quality (MAPQ) of 84.

No coverage IGV plot is presented.

We do not show coverage plots as that in our opinion is not useful to show. This is because we would have to show coverage plots for all samples and all included subgenomic references. However, the results from this mapping is shown in what is now Table 2 with more details in Supplementary Table S2. We chose to do the reads mapping and counting in IGV, as that allows both visual inspection as well as getting the reads counts, as given in Table 2. In addition, we have modified that sentence to instead state: “Abundance of mapped reads were determined in IGV at a minimal alignment score of 60 and a mapping quality (MAPQ) of 84” on page 24 of the revised manuscript.

6. It is recommended to provide the bam files available as well as the genome used to map. The bam files and the reference/s used had already been deposited in the NCBI SRA and have now been released so that the reviewer can assess it if needed. In addition, we have added Table 3 and Supplementary Table S2 giving additional details about number of reads

mapped to individual amplicons etc.

7. Authors write “56 million NGS reads generated from the 14 virus-positive samples, nearly 800,000 reads mapped to one of the 10 SARS-CoV-2 subgenomic RNAs”

In other words the whole analysis here is to 1.4% of the reads. Need to explain why is this number so low.

We have added several additional ways of comparing abundance of reads and believe that we have now covered the concerns raised here.

8. The number of reads that align the whole genome should be used for normalization between the samples. The way Authors performed normalization is not clear.

As mentioned above, we have provided more details and provide several additional ways of comparing abundance of reads and believe that we have now covered the concerns raised.

Results

1. Table 1 (and all other tables and plots) should contain counts to orf1a, orf1b and the human controls.

As mentioned above, we have added details and present different ways of comparing reads and PCR results.

2. Please explain what is this full virus genome? Looking at Authors previous article there are several genomes.

The full virus genome used throughout is the reference virus sequence given and not our own assembled genomes. Consequently, and we have tried to make this clear, the reference genome used is based on Wuhan-Hu-1-NC_045512/MN908947.3 only.

3. Supplementary Figure S5: Average coverage per 5 million reads for samples 37, 38, 60, 61, 62 and 63

This plot has repeated measures of two people yet it is not easy to follow which is a repeated measure of whom.

We have extensively revised or removed Figures and have tried to clarify the connection between samples and individuals, in particular by adding a new Table 1 with those details and have also made it more clear in other Tables.

4. In general the only plots presented are histograms. Yet, they are not a good choice since the authors try to demonstrate two issues here:

a. There is a difference within a certain sample between the expressions of the various regions.

b. There is difference between the samples “profile”, since some samples are more degraded or from a later stage in the disease.

This can be demonstrated using clustering among the samples and a heat map of standardized values.

We now present the main results in Tables and in box-and-whiskers plots. In the text, we present different ways of comparing the data as well.

5. Title - Detection of subgenomic RNAs mapped to the virus genome by filtering reads containing the partial leader sequence

The word 'the' should be emitted.

Thanks, but we are not sure exactly which of the words “the” the reviewer is referring to? As we see it, it should be both “the virus genome” as the mapping was done to the reference virus genome (we have added “reference” to make this clear), and also “the partial leader sequence” as it is the same partial leader sequence for the different subgenomic RNAs. We hope we have made this clear, in particular as to us having used the reference genome throughout, not our own assembled virus genomes.

6. Authors write “only barcodes” , this is the first mention of barcodes, perhaps authors mean samples.

Thanks for finding this mistake. The sample nomenclature has been updated throughout.

Discussion

1. Authors write “Two different approaches used”, need to explain what they are.

Details have been added to page 17 to make this clear, thanks.

2. Authors write “as we believe the study described by Zhang et al. is not mapping subgenomic RNAs but simply reports coverage for the different parts of the virus genome.” What do Authors mean?

Zhang et al., do not look for or map reads that contain the leader sequence common to all the subgenomic RNAs. Consequently, their mapping is simply mapping reads to their position in the genome without any distinction between leader-containing subgenomic RNAs or not. Consequently, their mapping is not comparable to our study as we specifically look for leader-containing subgenomic RNAs. We have clarified this in the revised manuscript on page 21.

3. Authors write; “we present information that helps understand prolonged and sometimes inconsistent PCR-positivity and may pave the way for development of better diagnostic PCRs”

How exactly are the authors proposing to improve the diagnosis and the PCR?

First of all, knowing that subgenomic RNAs may be rather stable may aid in avoiding using their presence or absence to state active replication/transcription or not. Secondly, understanding the presence and abundance of these RNAs, particular in poor quality or low virus load samples, may lead to development of particular PCRs, e.g. highly sensitive PCRs capable of detecting negative strand replicative intermediates, that may be better at distinguishing between early/active infection from later positive detection of highly protected RNA. Although we were not able to detect double stranded forms of the SARS-CoV-2 RNAs, these are likely present in early infection and as mentioned, would require more sensitive methods to detect. We have added text discussing this in the revised manuscript; however, based on our findings this far, it appears that ratios of virion to subgenomic RNAs, and the ratios of positive strand to negative strands, are somewhat constant/stable, and consequently, that our findings are more about understanding that

these RNAs are present and can be detected for an extended period and should not be taken as evidence of early/active infection as that assumption likely came from cell culture studies in which early/active infection is 6-8 hours after infection and consequently measured in hours and not days as for infection *in vivo*.

REVIEWER COMMENTS, second round -

Reviewer #1 (Remarks to the Author):

Thanks for including the experimental data and revising the manuscript extensively.

Reviewer #2 (Remarks to the Author):

The revised version still suffers from:

1. Excessive verbiage, difficult-to-follow text.
2. Difficult to interpret data almost entirely in the form of tables. Graphical display of experimental design and data would greatly improve things.

In the revised version authors have gone into great detail describing methods for identifying NGS reads corresponding to subgenomic RNAs from samples obtained from patients, and also compare these results to others' published data. The methods are much more clearly described than in the previous draft, especially in the Methods section. However this time, the text in the Results is exceedingly long. Lines 148-389 (6 pages at 1.5 line-spacing) is devoted to discussing different ways in which subgenomic RNAs were identified, how many were present for each subgenomic RNA in each patient, and how they compare to other databases. I recommend that this text be shortened by about 50%. Readers can glean much of the data from the tables, rather than authors repeating in the text. Plus the Results repeat text in the Methods. Also, making the paper hard to read is that almost all of the data are in large, complicated Tables. While that information is necessary, this reviewer re-iterates that diagrams showing sequence analysis strategies, with, for example, maps showing where reads map to TRS-subgenomic RNA junctions, or graphics showing graphs of number of reads beside subgenomic RNA maps, would make this paper much more readable, i.e. the authors use a lot of words to describe strategies and RNAs that would be more easily interpreted via diagrams. There are many software packages available for plotting NGS data in informative ways. A less complicated example of a graphic would be to convert the summary ratios of reads in Lines 274-378 to a graph.

Although the experiments are explained better, it is still difficult to remember the point of the paper., while one is reading it, because of the dense, abstruse writing that makes this paper so hard to follow. Throughout the paper the authors need to break up sentences to make them more readable. A sentence in the Abstract (lines 22-29 is seven lines long)! By the time the reader finishes the sentence they lose track of what the authors started out saying at the beginning. The same applies to the first sentence of the section on RT-PCR to detect subgenomic RNA 7a (lines 393-400). It contains a two line parenthetical phrase, two semicolons all to say that the results are in a Figure. This reviewer suggests the authors start the paragraph with a topic sentence stating what question is being asked next and why, and then list the PCR experiments that are shown in Table 4 in one sentence for each, or as a numbered list. Table 4 would be much easier to interpret if presented in graphical form.

The part on RNA in replication vesicles is improved, by showing susceptibility to RNase upon detergent treatment, which supports the statement in the current version of the abstract that this is evidence for "likely protection by cellular membranes consistent with being part of virus-induced replication organelles."

Overall, the most important aspect of this manuscript seems to be that subgenomic RNAs seem to be detected in patients at a time when no replication is occurring, and/or virion preparations may also include membrane-bound replication factories which would give the impression of subgenomic RNAs in virions. The amount of text used to convey this point is excessive. There is such detailed discussion of the peculiarities of subgenomic RNA levels in each patient that the take-home lesson in the title of the paper gets lost. As yet another example of this, the Discussion, which simply repeats much of the Results has three paragraphs that begin "In conclusion," The Discussion could be cut in half, with final paragraph being the only "In conclusion" paragraph because it makes the most important point of the paper that subgenomic RNAs are disproportionately favored in degraded samples and at time points well after replication is thought to be concluded. Thus,

clinicians should not conclude virus is still replicating, just because subgenomic RNAs are detected.

I am not sure if this work, consisting entirely of deep sequencing data is really of sufficient general interest or significance to its field for publication in Nature Communications, rather than a more specialized virology or clinical journal. I'll leave that up to the editor.

Reviewer #3 (Remarks to the Author):

In general, the authors made many important changes in this revision; they added PCR validations and made major changes in the analysis. I describe here additional required changes.

1) The authors added the quantification of the cellular RNA (as I suggested) and this helps monitor bad quality samples.

Since in general what we have are three categories of amplicons: cellular, first 21000 bases of the virus and the subgenomic regions, it would be easier to follow a table with four numbers per sample:

a) Reads mapped to first 21,000

b) Reads mapped to all subgenomic regions

c) Reads mapped to cellular RNA

d) Reads not mapped to any of the above (with sufficient confidence)

The sum of a-d should be total amount of reads

Using this approach the four numbers above can be presented as a pie chart per sample. It will easily demonstrate which are the "bad" samples as well as the variation in the distributions between the categories.

2) The authors replaced the histograms (as I suggested) to a box plot - Figure 1.

I find this helpful, yet the outliers of orf7a reach 200,000 reads and make all the box plots presented too condensed. Therefore, I suggest restricting the y-axis to ~60,000

3) How read quantification of the amplicons was performed is still not clear to me. The authors write:

"Abundance of mapped reads were determined in IGV at a minimal alignment score of 60 and a mapping quality (MAPQ) of 84"

IGV is genome browser. It contains igvtools, did they use that? To the best of my knowledge, this tool will not quantify per amplicon. In addition within the bam files there is a tag - XS <http://129.130.90.13/ion-docs/GUID-965C5ED4-20C8-45D5-AF07-8B0008AF74AD.html>, its definition is - "The alignment score of next-best sub-optimal mapping". Therefore, please clarify what TAG was used for selecting "a minimal alignment score of 60".

4) In the result section- Comparison of abundance of reads mapped to virus and cellular amplicons and abundance of reads mapped specifically to subgenomic RNAs, I would appreciate having a table to summarize the various ways of calculation and their result.

5) What are 'Artic network primers' written in results?

REVIEWER COMMENTS

Reviewer #1 (Remarks to the Author):

Thanks for including the experimental data and revising the manuscript extensively.

We are very pleased with this support from the reviewer.

Reviewer #2 (Remarks to the Author):

The revised version still suffers from:

1. Excessive verbiage, difficult-to-follow text.

We have gone through the text and revised it to make it shorter and easier to follow as detailed in more details below.

2. Difficult to interpret data almost entirely in the form of tables. Graphical display of experimental design and data would greatly improve things.

We have changed and added Figures to display some of the data in addition to Tables. Consequently, we now have 4 Figures in the main text and 5 Figures in the Supplementary Information.

In the revised version authors have gone into great detail describing methods for identifying NGS reads corresponding to subgenomic RNAs from samples obtained from patients, and also compare these results to others' published data. The methods are much more clearly described than in the previous draft, especially in the Methods section. However this time, the text in the Results is exceedingly long. Lines 148-389 (6 pages at 1.5 line-spacing) is devoted to discussing different ways in which subgenomic RNAs were identified, how many were present for each subgenomic RNA in each patient, and how they compare to other databases. I recommend that this text be shortened by about 50%. Readers can glean much of the data from the tables, rather than authors repeating in the text. Plus the Results repeat text in the Methods. Also, making the paper hard to read is that almost all of the data are in large, complicated Tables. While that information is necessary, this reviewer re-iterates that diagrams showing sequence analysis strategies, with, for example, maps showing where reads map to TRS-subgenomic RNA junctions, or graphics showing graphs of number of reads beside subgenomic RNA maps, would make this paper much more readable, i.e. the authors use a lot of words to describe strategies and RNAs that would be more easily interpreted via diagrams. There are many software packages available for plotting NGS data in informative ways. A less complicated example of a graphic would be to convert the summary ratios of reads in Lines 274-378 to a graph.

The indicated parts of the document, as well as other parts, have been revised to make it shorter and easier to follow. In addition, we have updated the previous Figure 1 to now also show the structure of the SARS-CoV-2 genome as well as the subgenomic RNAs together with a violin plot of the number of reads mapped to the individual subgenomic RNAs. The

text around the ratios on what was lines 274-378 previously has also been shortened and a new Figure 2 and Figure 3, as well as Supplementary Figures 2 and 3, added to show the data. Additional details/numbers are now also included in the Source Data file.

Although the experiments are explained better, it is still difficult to remember the point of the paper., while one is reading it, because of the dense, abstruse writing that makes this paper so hard to follow. Throughout the paper the authors need to break up sentences to make them more readable. A sentence in the Abstract (lines 22-29 is seven lines long)! By the time the reader finishes the sentence they lose track of what the authors started out saying at the beginning. The same applies to the first sentence of the section on RT-PCR to detect subgenomic RNA 7a (lines 393-400). It contains a two line parenthetical phrase, two semicolons all to say that the results are in a Figure. This reviewer suggests the authors start the paragraph with a topic sentence stating what question is being asked next and why, and then list the PCR experiments that are shown in Table 4 in one sentence for each, or as a numbered list. Table 4 would be much easier to interpret if presented in graphical form.

The indicated parts of the document, as well as other parts, have been revised to make it shorter and easier to follow. In addition, we have added a new Figure 4 and a Supplementary Figure 4 to show the data.

The part on RNA in replication vesicles is improved, by showing susceptibility to RNase upon detergent treatment, which supports the statement in the current version of the abstract that this is evidence for “likely protection by cellular membranes consistent with being part of virus-induced replication organelles.”

Thank you for these positive comments. To further improve the presentation of this section of the results, we have added a new Supplementary Figure 5.

Overall, the most important aspect of this manuscript seems to be that subgenomic RNAs seem to be detected in patients at a time when no replication is occurring, and/or virion preparations may also include membrane-bound replication factories which would give the impression of subgenomic RNAs in virions. The amount of text used to convey this point is excessive. There is such detailed discussion of the peculiarities of subgenomic RNA levels in each patient that the take-home lesson in the title of the paper gets lost. As yet another example of this, the Discussion, which simply repeats much of the Results has three paragraphs that begin “In conclusion,” The Discussion could be cut in half, with final paragraph being the only “In conclusion” paragraph because it makes the most important point of the paper that subgenomic RNAs are disproportionately favored in degraded samples and at time points well after replication is thought to be concluded. Thus, clinicians should not conclude virus is still replicating, just because subgenomic RNAs are detected.

The indicated parts of the document, as well as other parts, have been revised to make it shorter and easier to follow.

I am not sure if this work, consisting entirely of deep sequencing data is really of sufficient general interest or significance to its field for publication in Nature Communications, rather than a more specialized virology or clinical journal. I'll leave that up to the editor.

We do not agree with this comment from the reviewer as our revised version, including the previous one, is not consisting entirely of deep sequencing data, but now rather combine the sequencing data with PCR data as well as fractionation data to look at potential membrane association. We think the finding is of importance as we show that reads mapping to coronavirus subgenomic RNAs are present in most diagnostic swab samples even relatively long after first detection, and therefore are not a good indicator of active/recent replication/transcription. This is followed up by showing that this finding can be supported by PCR results as well as membrane protection experiments. Finally, we show that such reads mapping to coronavirus subgenomic RNAs can also be found in deposited sequence reads archives.

Reviewer #3 (Remarks to the Author):

In general, the authors made many important changes in this revision; they added PCR validations and made major changes in the analysis. I describe here additional required changes.

1) The authors added the quantification of the cellular RNA (as I suggested) and this helps monitor bad quality samples.

Since in general what we have are three categories of amplicons: cellular, first 21000 bases of the virus and the subgenomic regions, it would be easier to follow a table with four numbers per sample:

- a) Reads mapped to first 21,000
- b) Reads mapped to all subgenomic regions
- c) Reads mapped to cellular RNA
- d) Reads not mapped to any of the above (with sufficient confidence)

The sum of a-d should be total amount of reads

Using this approach the four numbers above can be presented as a pie chart per sample. It will easily demonstrate which are the "bad" samples as well as the variation in the distributions between the categories.

We thank the reviewer for this suggestion and have added a new Figure 2 with this information.

2) The authors replaced the histograms (as I suggested) to a box plot - Figure 1. I find this helpful, yet the outliers of orf7a reach 200,000 reads and make all the box plots presented too condensed. Therefore, I suggest restricting the y-axis to ~60,000

Thank you for the comment. We have changed Figure 1, as well as Supplementary Figure 1, to violin plots that better show the full distribution of counts in the samples.

3) How read quantification of the amplicons was performed is still not clear to me. The authors write:

“Abundance of mapped reads were determined in IGV at a minimal alignment score of 60 and a mapping quality (MAPQ) of 84”

IGV is genome browser. It contains igvtools, did they use that? To the best of my knowledge, this tool will not quantify per amplicon. In addition within the bam files there is a tag – XS <http://129.130.90.13/ion-docs/GUID-965C5ED4-20C8-45D5-AF07-8B0008AF74AD.html>, its definition is – “The alignment score of next-best sub-optimal mapping”. Therefore, please clarify what TAG was used for selecting “a minimal alignment score of 60”.

We have added more explanation to the Materials and Methods section so it now reads “Mapped reads were visualised in IGV at a minimal alignment score of 60 and a mapping quality (MAPQ) of 84 and the abundance of reads mapped specifically to each subgenomic RNA at this stringency assessed by recording the read coverage at nucleotide position 61 of the leader sequence.” Together with the information now given in the new Figure 1, together with information in the fasta file available in the Supplementary Information and on the NCBI SRA, we hope that the strategy for mapping is now clear.

4) In the result section– Comparison of abundance of reads mapped to virus and cellular amplicons and abundance of reads mapped specifically to subgenomic RNAs, I would appreciate having a table to summarize the various ways of calculation and their result.

As mentioned above in response to Reviewer 2, we have revised and shortened the text around the ratios on what was lines 274-378 previously has also been shortened and a new Figure 2 and Figure 3, as well as Supplementary Figures 2 and 3, added to show the data. Additional details/numbers are now also included in the Source Data file.

5) What are ‘Artic network primers’ written in results?

We should of course had made this clear and have now added a reference with the details and link to the protocol (Quick, J. nCoV-2019 sequencing protocol v2 (GunIt). *protocols.io*, doi:<https://dx.doi.org/10.17504/protocols.io.bdp7i5rn> (2020)).

I look forward to hearing back.

Sincerely,

Prof. Soren Alexandersen, DVM, PhD, DVSc., FRCPath, MRCVS
Director, Geelong Centre for Emerging Infectious Diseases
Deakin University, School of Medicine, Faculty of Health
Health Education and Research Building (HERB) Level 3, University Hospital Geelong, 285
Ryrie Street, PO Box 281 Geelong VIC 3220, Australia
E-mail: Soren.Alexandersen@deakin.edu.au Mobile: +61 (0)4 27282311